# CONSTRAINED MULTI-OBJECTIVE REINFORCEMENT LEARNING WITH MAX-MIN CRITERION

## ABSTRACT

Multi-Objective Reinforcement Learning (MORL) extends standard RL by optimizing policies over multiple and often conflicting objectives. Although max-min scalarization has emerged as a powerful approach to promote fairness in MORL, it has limited applicability, especially when incorporating constraints. In this paper, we propose a unified framework for constrained MORL that combines the max-min criterion with constraint satisfaction and generalizes prior formulations such as unconstrained max-min MORL and constrained weighted-sum MORL. We establish a theoretical foundation for our framework and validate our algorithm through a formal convergence analysis and experiments in tabular environments. We extend our framework to practical applications, including simulated edge computing resource allocation and locomotion control. Across these domains, the method demonstrates strong handling of fairness and constraint satisfaction in multi-objective decision-making.

## 1 INTRODUCTION

Reinforcement Learning (RL) is a powerful machine learning framework that enables an agent to learn optimal decision-making strategies through interaction with an environment. In recent years, Multi-Objective Reinforcement Learning (MORL) has gained significant interest because many real-world control problems inherently involve multiple, often conflicting objectives (Roijers et al., 2013; Yang et al., 2019; Hayes et al., 2022; Basaklar et al., 2023; Park et al., 2024; Park & Sung, 2025). MORL extends standard RL to handle simultaneous optimization of multiple objectives.

A common strategy in MORL involves optimizing a scalarized function defined over multiple objective returns (Roijers et al., 2013; Hayes et al., 2022). This framework seeks to identify a policy $\pi$ that maximizes a scalarized value $f(J_1(\pi), \ldots, J_K(\pi))$, where each $J_k(\pi)$ represents the expected discounted return for the $k$-th objective among $K(\geq 2)$ objectives, and $f : \mathbb{R}^K \to \mathbb{R}$ is a non-decreasing scalarization function such that $J_k(\pi) \geq J_k(\pi'), 1 \leq k \leq K \Rightarrow f(J(\pi)) \geq f(J(\pi'))$. Thus, $f$ plays a key role in imposing the designer's preference among multiple objectives.

Although much of the MORL literature employs a linear $f$ (that is, the weighted sum: $\max_\pi \sum_{k=1}^K w_k J_k(\pi)$) due to its simplicity, the weighted sum does not always accurately represent the preference of a designer, especially regarding fairness among objectives (Hayes et al., 2022; Park et al., 2024). For instance, imagine a traffic light system managing an intersection where several roads converge with asymmetric arrival rates. Instead of simply aiming to reduce the total sum waiting time for all vehicles across the roads, the designer could prioritize fairness by minimizing the longest individual waiting time among the roads. This helps reduce localized congestion (Raeis & Leon-Garcia, 2021) and avoid severe delays for individual drivers.

Fairness-driven objectives frequently arise in real-world scenarios and are addressed using scalarization methods beyond the standard weighted sum, such as max-min optimization or proportionally fair optimization (Khan et al., 2016) in MORL. While proportionally fair optimization, expressed as $\max_\pi \sum_{k=1}^K w_k \log J_k(\pi)$, is relatively straightforward to solve due to the smoothness and differentiability of the $\log$ function, max-min optimization presents greater challenges because of its non-differentiability and non-linearity. Recently, Park et al. (2024) proposed an algorithm to explicitly address the max-min objective in MORL using Gaussian smoothing (Nesterov & Spokoiny, 2017).

Although max-min optimization in MORL is a powerful tool with broad applicability (Regan & Boutilier, 2010; Zehavi et al., 2013; Saifullah et al., 2014; Wang et al., 2019; Chakraborty et al., 2024) such as mitigating bottlenecks in cloud and edge resource management systems (Saifullah et al., 2014; Wang et al., 2019), the standard framework lacks flexibility for diverse problem types. First, it is designed to ensure fairness across homogeneous objectives, but applying max-min fairness to heterogeneous objectives, such as velocity and energy consumption in locomotion, is inappropriate due to their differing units and nature. In our context, two physical quantities are considered heterogeneous if they have different units. In such cases, one may maximize the minimum of homogeneous objectives while requiring other objectives to remain above certain thresholds. Second, many real-world problems involve constraints that must be satisfied. For example, in resource allocation, a MORL-based scheduler may aim to maximize throughput and fairness across task queues under a strict power consumption constraint (Chen et al., 2020; Jiang et al., 2020). Incorporating constraints into the max-min MORL framework thus significantly broadens its practical applicability.

In this paper, we propose a novel framework for constrained MORL that incorporates max-min fairness. Our approach is capable of maximizing the max-min fairness among homogeneous objectives while simultaneously incorporating other heterogeneous quantities as constraints. We present a detailed theoretical basis for our algorithmic design. Moreover, our framework generalizes previous frameworks in MORL, including the original max-min MORL formulation (Park et al., 2024) and constrained weighted-sum MORL (Huang et al., 2021). Our main contributions are as follows:

• We introduce a unified framework for constrained MORL that integrates the max-min criterion and establishes its theoretical foundations, including differentiability, twice-differentiability, and smoothness of our objective function.

• We propose an iterative algorithm for constrained max-min MORL, accompanied by a formal convergence analysis. We empirically assess its convergence in tabular environments.

• We establish the practical relevance of our method by applying it to simulated edge computing resource allocation and locomotion control. Across these scenarios, our method demonstrates its ability to better balance max-min fairness and constraint satisfaction than the considered baselines.

## 2 BACKGROUND

A multi-objective Markov decision process (MOMDP) is represented as $\langle \mathcal{S}, \mathcal{A}, T, \mu_0, r, \gamma \rangle$, where $\mathcal{S}$ and $\mathcal{A}$ are the sets of states and actions, respectively, $T$ represents the transition probability distribution, $\mu_0$ specifies the initial state distribution, and $\gamma \in [0, 1)$ is the discount factor. The reward function $r : \mathcal{S} \times \mathcal{A} \to \mathbb{R}^{K+L}$, $K \geq 1, L \geq 0$ is vector-valued with its $k$-th element denoted by $r^{(k)}$ $(1 \leq k \leq K + L)$ such that $|r^{(k)}| \leq r_{\max}^{(k)}$, where $K + L$ is the total number of objectives. At each timestep, the agent selects an action $a$ in the current state $s$ according to its (stationary) policy $\pi : \mathcal{S} \to \mathcal{P}(\mathcal{A})$, where $\mathcal{P}(\mathcal{A})$ represents the set of probability distributions in the action space $\mathcal{A}$. The occupancy measure is defined as $\rho(s, a) := \sum_{s'} \mu_0(s') \sum_{t=0}^{\infty} \gamma^t \Pr(s_t = s, a_t = a | s_0 = s', \pi^\rho)$ where $\pi^\rho$ is the corresponding stationary policy induced by $\rho$, expressed as $\pi^\rho(a|s) = \frac{\rho(s,a)}{\sum_{a'} \rho(s,a')}$ (Puterman, 1994). Then, the vector return evaluated by $\pi^\rho$ is given by

$$J(\pi^\rho) := [J_1(\pi^\rho), \cdots, J_{K+L}(\pi^\rho)]^\top = \mathbb{E}_{\pi^\rho}\left[\sum_{t=0}^{\infty} \gamma^t r_t\right] = \sum_{(s,a)} r(s,a)\rho(s,a) \in \mathbb{R}^{K+L}. \quad (1)$$

## 3 CONSTRAINED MAX-MIN MORL FRAMEWORK

### 3.1 THEORETICAL FOUNDATION

We consider constrained MORL, where the last $L$ of the total $K + L$ objectives should satisfy certain constraints. For theoretical development in this section, we assume that $\mathcal{S}$ and $\mathcal{A}$ are finite. The problem is formulated as follows:

$$\max_{\pi^\rho} \ f(J_1(\pi^\rho), \cdots, J_K(\pi^\rho)) + \beta \sum_s \mathcal{H}_\rho(s)\rho(s) \quad (2)$$

$$\text{s.t.} \ J_{K+l}(\pi^\rho) \geq C^{(l)}, \ l = 1, \cdots, L \quad (3)$$

where $\mathcal{H}_\rho(s) := -\sum_a \pi^\rho(a|s) \log \pi^\rho(a|s)$ is the entropy of $\pi^\rho(\cdot|s)$, $\rho(s) := \sum_a \rho(s,a)$ is the stationary state distribution in $\mathcal{S}$, $\beta > 0$ is a balancing coefficient, and $\{C^{(l)}\}_{l=1}^L$ is a set of threshold values. We assume a mild condition that the set $\{C^{(l)}\}_{l=1}^L$ is chosen by the designer such that the optimization in equation 2 and equation 3 is feasible, an assumption commonly made in the constrained MDP literature (Tessler et al., 2018; Ha et al., 2020).

In this paper, we set $f$ the minimum function, i.e., $f(J_1(\pi^\rho), \cdots, J_K(\pi^\rho)) = \min(J_1(\pi^\rho), \cdots, J_K(\pi^\rho))$. We note that the entropy term is included in equation 2 to promote exploration and eliminate the indeterminacy of the max-min solution without the entropy term (Park et al., 2024). The problem reduces to the unregularized formulation as $\beta \to 0$, with the optimality gap decreasing linearly:

**Proposition 3.1.** *The gap between the optimal max-min value of the unregularized problem and that of the regularized problem in equation 2 and equation 3 with $f = \min$ is upper bounded by $\frac{\beta \log |\mathcal{A}|}{1-\gamma}$.* *(Proof: See Appendix A.)*

Proposition 3.1 shows that the regularized problem is a valid approximation of the unregularized criterion. Since directly optimizing equation 2 and equation 3 with $f = \min$ and $J_k(\pi^\rho) = \mathbb{E}_{\pi^\rho}[\sum_{t=0}^\infty \gamma^t r_t^{(k)}]$ is non-trivial due to its non-differentiable and non-linear structure, we address this challenge using the occupancy measure (i.e., stationary distribution (Puterman, 1994)) formulation. The above optimization problem with $f = \min$ can be rewritten as

$$\max_{\rho \geq 0} \min_{1 \leq k \leq K} \left( \sum_{(s,a)} r^{(k)}(s,a)\rho(s,a) \right) + \beta \sum_s \mathcal{H}_\rho(s)\rho(s) \tag{4}$$

$$\sum_{a'} \rho(s',a') = \mu_0(s') + \gamma \sum_{(s,a)} T(s'|s,a)\rho(s,a), \ \forall s' \tag{5}$$

$$\sum_{(s,a)} c^{(l)}(s,a)\rho(s,a) \geq C^{(l)}, \ l = 1, \cdots, L \tag{6}$$

where equation 5 is the Bellman flow equation for the occupancy measure (Puterman, 1994). Here, we use the notation $c^{(l)}(s,a) := r^{(K+l)}(s,a), \ l = 1, \cdots, L$ to explicitly represent the dimensions associated with the constraint. These quantities can be true rewards or negative of costs. Then the formulation in equation 4, equation 5, and equation 6 constitutes a convex optimization problem. Now we derive a convex optimization equivalent to the dual problem of equation 4, equation 5, and equation 6, which serves as the foundation for our subsequent model-free applications (Section 5.2), as stated in the following proposition.

**Proposition 3.2.** *The dual problem of equation 4, equation 5, and equation 6 is equivalent to the following convex optimization problem:*

$$\min_{u \in \mathbb{R}_+^L, w \in \Delta^K} \mathcal{L}(u,w) = \sum_s \mu_0(s)v_{u,w}^*(s) - \sum_{l=1}^L u_l C^{(l)} \tag{7}$$

*where $\mathbb{R}_+^L := \{u \in \mathbb{R}^L | u_l \geq 0, \ 1 \leq l \leq L\}$, $\Delta^K := \{w \in \mathbb{R}^K | \sum_{k=1}^K w_k = 1; \ w_k \geq 0, \ 1 \leq k \leq K\}$, i.e., the $(K-1)$-dimensional simplex, and $v_{u,w}^*$ is the fixed point of the operator $\mathcal{T}_{u,w}$:*

$$[\mathcal{T}_{u,w}v](s) = \beta \log \sum_a \exp[\frac{1}{\beta}\{\sum_{l=1}^L u_l c^{(l)}(s,a) + \sum_{k=1}^K w_k r^{(k)}(s,a) + \gamma \sum_{s'} T(s'|s,a)v(s')\}], \ \forall s. \tag{8}$$

*(Proof: See Appendix B.)*

Strong duality holds if there exists an occupancy measure $\rho$ such that $\rho(s,a) > 0, \ \forall(s,a)$ and the constraints in equation 6 are satisfied with strict inequalities, assumptions commonly used in RL (Lee et al., 2021) and constrained RL settings (Tessler et al., 2018; Ha et al., 2020).

Proposition 3.2 hints that $v_{u,w}^*$ can be obtained via soft value iteration in equation 8 and the weights $u$ and $w$ can be obtained by minimizing the loss $\mathcal{L}(u,w)$ in equation 7 by some method. In addition, in equation 8, we observe that the constrained reward $c^{(l)}, \ l = 1, \cdots, L$ can be handled without

distinction from the unconstrained reward $r^{(k)}$, $k = 1, \cdots, K$. Note that both rewards appear as a weighted sum in equation 8, enabling a unified framework for constrained and unconstrained MORL.

However, solving the optimization problem equation 7 directly is non-trivial because the fixed point $v^*_{u,w}$ in equation 8 does not have a closed-form expression in terms of $(u, w)$. To address this issue, we derive the key properties of $v^*_{u,w}$. For given $(u, w)$, we define

$$Q^*_{u,w}(s, a) := \sum_{l=1}^{L} u_l c^{(l)}(s, a) + \sum_{k=1}^{K} w_k r^{(k)}(s, a) + \gamma \sum_{s'} T(s'|s, a) v^*_{u,w}(s'), \tag{9}$$

and define a policy $\pi^*_{u,w}$ as

$$\pi^*_{u,w}(a|s) = \frac{\exp(\frac{1}{\beta} Q^*_{u,w}(s, a))}{\sum_{a'} \exp(\frac{1}{\beta} Q^*_{u,w}(s, a'))}. \tag{10}$$

Then, $\pi^*_{u,w}$ is an optimal policy for the entropy-regularized RL (Haarnoja et al., 2017) with a scalar reward function $\sum_{l=1}^{L} u_l c^{(l)}(s, a) + \sum_{k=1}^{K} w_k r^{(k)}(s, a)$. Furthermore, regarding the relationship between $\pi^*_{u,w}$ and the gradient of $v^*_{u,w}$, we have the following theorem:

**Theorem 3.3.** *For each $s$, $v^*_{u,w}(s)$ is differentiable w.r.t. $(u, w) \in \mathbb{R}^{L+K}$, and its gradient $\nabla v^*_{u,w}(s) = [\nabla_u v^*_{u,w}(s)^\top, \nabla_w v^*_{u,w}(s)^\top]^\top$ has the form of*

$$\nabla_u v^*_{u,w}(s) = v_c^{\pi^*_{u,w}}(s) \ \text{and} \ \nabla_w v^*_{u,w}(s) = v_r^{\pi^*_{u,w}}(s), \tag{11}$$

*where $v_c^{\pi^*_{u,w}}(s) \in \mathbb{R}^L$ and $v_r^{\pi^*_{u,w}}(s) \in \mathbb{R}^K$ are the value functions evaluated with the policy $\pi^*_{u,w}$ for the constrained reward $c^{(l)}$ and the unconstrained reward $r^{(k)}$, respectively. (Proof: See Appendix C.)*

Theorem 3.3 implies that the objective function $\mathcal{L}(u, w)$ in equation 7 is differentiable with respect to (w.r.t.) $(u, w)$, and enables us to apply gradient descent to solve the optimization with the gradient $(\nabla_v v^*_{u,w}(s), \nabla_w v^*_{u,w}(s))$ combined with value iteration.

It is surprising but makes sense that the gradient $\nabla v^*_{u,w}(s)$ is expressed as the value function (which is a vector quantity) evaluated with the policy $\pi^*_{u,w}$. First, consider the constrained part. Due to Theorem 3.3, the derivative of $\mathcal{L}(u, w)$ in equation 7 is given by $\sum_s \mu_0(s) v_c^{\pi^*_{u,w}}(s) - [C^{(1)}, \cdots, C^{(L)}]^\top$. Hence, if the value of the $l$-th constrained dimension is larger than $C^{(l)}$, then the $l$-th component of the gradient is positive, gradient descent will decrease the weight $u_l$, and hence $c^{(l)}$ is less weighted in the value iteration in equation 8. Otherwise, the opposite happens. In this way, the constraints on the constrained dimensions are satisfied with gradient descent.

Regarding the unconstrained reward part, the gradient is given by $\sum_s \mu_0(s) v_r^{\pi^*_{u,w}}(s)$. Hence, for the dimension of a smaller value, we have a smaller reduction in $w_k$ by gradient descent to yield a larger $w_k$. Therefore, the dimensions with smaller values are weighted more in the value iteration in equation 8 to realize the max-min principle.

We now establish the twice-differentiability of $v^*_{u,w}$ to derive its Hessian. This step is crucial for establishing the smoothness of the objective function, which in turn is critical for analyzing the convergence of our algorithm in Section 3.2.

**Theorem 3.4.** *For each $s$, $v^*_{u,w}(s)$ is twice-differentiable w.r.t. $(u, w) \in \mathbb{R}^{L+K}$. Let $|\mathcal{S}| = p$, and suppose the states are enumerated as $\{s_1, \cdots, s_p\}$. Then, the $(L + K) \times (L + K)$ Hessian matrix $H[v^*_{u,w}(s_k)]$, $1 \le k \le p$, has the form of*

$$H[v^*_{u,w}(s_k)] = \frac{1}{\beta} \sum_{l=1}^{p} [(I_p - \gamma T^{\pi^*_{u,w}})^{-1}]_{kl} B^{\pi^*_{u,w}}(s_l). \tag{12}$$

*Here, $I_p$ is the $p \times p$ identity matrix; $T^{\pi^*_{u,w}}$ is a $p \times p$ matrix of which $i$-th row and $j$-th column element is given by $[T^{\pi^*_{u,w}}]_{ij} = \mathbb{E}_{a \sim \pi^*_{u,w}(\cdot|s_i)}[T(s_j|s_i, a)]$ $(1 \le i, j \le p)$; $[(I_p - \gamma T^{\pi^*_{u,w}})^{-1}]_{kl}$ denotes the $k$-th row and $l$-th column element of $(I_p - \gamma T^{\pi^*_{u,w}})^{-1}$; $B^{\pi^*_{u,w}}(s) = \mathbb{E}_{a \sim \pi^*_{u,w}(\cdot|s)}\Big[(Q^{\pi^*_{u,w}}(s, a) - $*

$\mathbb{E}_{a' \sim \pi_{u,w}^*(\cdot|s)}[Q^{\pi_{u,w}^*}(s,a')])(Q^{\pi_{u,w}^*}(s,a) - \mathbb{E}_{a' \sim \pi_{u,w}^*(\cdot|s)}[Q^{\pi_{u,w}^*}(s,a')])^\top] \in \mathbb{R}^{(L+K)\times(L+K)}$; and

$Q^{\pi_{u,w}^*}(s,a) \in \mathbb{R}^{L+K}$ is the value function evaluated with the policy $\pi_{u,w}^*$. (Proof: See Appendix D.)

Due to Theorem 3.4, the objective function $\mathcal{L}(u,w)$ in equation 7 is twice-differentiable w.r.t. $(u,w)$. Note that $Q^{\pi_{u,w}^*}(s,a)$ in Theorem 3.4 is different from $Q_{u,w}^*(s,a)$ in equation 9. By definition in the entropy-regularized RL, $Q_{u,w}^*(s,a) \in \mathbb{R}$ is the cumulative scalarized return plus the cumulative entropy sum from $\pi_{u,w}^*$. On the other hand, $Q^{\pi_{u,w}^*}(s,a) \in \mathbb{R}^{L+K}$ is a vector-valued cumulative sum of unconstrained rewards and constrained rewards from $\pi_{u,w}^*$ without the entropy sum. Therefore, $[u;w]^\top Q^{\pi_{u,w}^*}(s,a)$ equals to $Q_{u,w}^*(s,a)$ minus the cumulative entropy sum of $\pi_{u,w}^*$.

A natural approach to solving the convex optimization problem in equation 7 is projected gradient descent, since the variables $(u,w)$ lie in the convex set $\mathbb{R}_+^L \times \Delta^K$. The convergence of projected gradient descent depends on the smoothness of the objective function (Boyd & Vandenberghe, 2004; Bubeck, 2015). In our case, $\mathcal{L}(u,w)$ satisfies the following smoothness property:

**Theorem 3.5.** *For each $s$, $v_{u,w}^*(s)$ is smooth w.r.t. $(u,w)$ on $\mathbb{R}^{L+K}$. In other words, $\nabla v_{u,w}^*(s)$ is Lipschitz continuous in $\|\cdot\|_2$. Furthermore, $\mathcal{L}(u,w)$ is $\alpha$-smooth w.r.t. $(u,w)$ on $\mathbb{R}^{L+K}$ with $\alpha := \frac{1}{\beta(1-\gamma)}\sum_{m=1}^{L+K}\left(\frac{r_{max}^{(m)}}{1-\gamma}\right)^2$. (Proof: See Appendix E.)*

### 3.2 Algorithm and Convergence Analysis

Based on the foundation built in the previous section, we propose an algorithm for constrained MORL with max-min fairness. Note that we need to jointly update the weights $(u,w)$ and the value function, which approximates $v_{u,w}^*$. We adopt the following update method alternating between update of the value function and the weights $(u,w)$.

First, given a weight $(u,w)$, we update the value function to realize equation 8. For this, we use an action value function $Q$, which approximates $Q_{u,w}^*$. Using the soft Bellman equation (Haarnoja et al., 2017), the action value function $Q_{u,w}^*$ in equation 9 is written as $Q_{u,w}^*(s,a) = \sum_{l=1}^L u_l c^{(l)}(s,a) + \sum_{k=1}^K w_k r^{(k)}(s,a) + \gamma\sum_{s'}T(s'|s,a)v_{u,w}^*(s'), \forall(s,a)$. If we plug this equation into the right-hand side of equation 8, we have $v_{u,w}^*(s) = [\mathcal{T}_{u,w}v_{u,w}^*](s) = \beta\log\sum_a \exp\left(\frac{Q_{u,w}^*(s,a)}{\beta}\right)$ for each $s$. Using this form of $v_{u,w}^*(s)$, we implement applying $\mathcal{T}_{u,w}$ as updating the Q-function with the following:

$$Q(s,a) \leftarrow [u;w]^\top[c;r] + \gamma\sum_{s'}T(s'|s,a)\beta\log\sum_{a'}\exp\left(\frac{Q(s',a')}{\beta}\right), \forall(s,a). \tag{13}$$

We have shown that $\nabla_u v_{u,w}^*(s) = v_c^{\pi_{u,w}^*}(s)$, $\nabla_w v_{u,w}^*(s) = v_r^{\pi_{u,w}^*}(s)$ for each $s$, where we denote $v_c^{\pi_{u,w}^*}(s) \in \mathbb{R}^L$, $v_r^{\pi_{u,w}^*}(s) \in \mathbb{R}^K$ as the value functions evaluated with the policy $\pi_{u,w}^*$ for constrained reward $c$ and unconstrained reward $r$, respectively. We compute an estimated gradient of $\nabla_{(u,w)}\mathcal{L}(u,w)$ at the current weight $(u,w) = (u^m, w^m)$ where $m = 1, 2, \cdots$ is the iteration index. Note that the policy is extracted from the Q-function based on the form equation 10. We then update $(u,w)$ using projected gradient descent:

$$(u^{m+1}, w^{m+1}) = \mathcal{P}_{K,L}[(u^m, w^m) - l_w\nabla_{(u,w)}\mathcal{L}(u^m, w^m)] \tag{14}$$

where $l_w$ is a learning rate for $(u,w)$ and $\mathcal{P}_{K,L}[\cdot]$ is the projection onto the $\mathbb{R}_+^L \times \Delta^K$. We use the convex optimization method from Wang & Carreira-Perpiñán (2013) to project onto the simplex $\Delta^K$, and apply non-negativity clipping for projection onto $\mathbb{R}_+^L$. Note that the projection onto $\Delta^K$ is numerically stable as it is fully deterministic and avoids randomized procedures. In addition, its complexity is $O(K\log K)$ (Wang & Carreira-Perpiñán, 2013) which is relatively lightweight compared to other components, due to the sublinear growth of the logarithmic term.

We iterate this process for each $m$, and the pseudocode of our algorithm is shown in Algorithm 1. We now provide our convergence analysis of Algorithm 1 under the following assumption.

*Assumption* There exists at least one state $s \in \mathcal{S}$ such that the centered action-value vectors in the set $S_{\text{center}}(s) := \left\{Q^{\pi_{u,w}^*}(s,a) - \mathbb{E}_{a' \sim \pi_{u,w}^*(\cdot|s)}[Q^{\pi_{u,w}^*}(s,a')] : a \in \mathcal{A}\right\}$ span $\mathbb{R}^{K+L}$.

---

**Algorithm 1** Constrained Max-Min MORL Algorithm

---

1: $Q^0 \in \mathbb{R}^{|\mathcal{S}||\mathcal{A}|}$: initialized Q-function, ITER: total iteration number, $l_{\mathrm{w}}$: learning rate for the update of weights $(u, w)$
2: Initialize weights $u^0 \in \mathbb{R}_+^L$ and $w^0 \in \Delta^K$.
3: **for** $m = 1, 2, \cdots, \text{ITER}$ **do**
4:     $Q = Q^{m-1}$
5:     **while** not terminated **do**
6:         Update $Q$ in equation 13 with $[u; w] = [u^m; w^m]$.
7:     **end while**
8:     $Q^m = Q$
9:     Compute $\tilde{\nabla}_{(u,w)}\mathcal{L}(u^m, v^m)$, an estimated gradient of $\nabla_{(u,w)}\mathcal{L}(u^m, w^m)$ using $\pi^m(\cdot|s) = \mathrm{softmax}\{Q^m(s, \cdot)/\beta\}$ based on equation 11.
10:     $(u^{m+1}, w^{m+1}) = \mathcal{P}_{K,L}[(u^m, w^m) - l_{\mathrm{w}}\tilde{\nabla}_{(u,w)}\mathcal{L}(u^m, w^m)]$.
11: **end for**
12: Return $\pi(\cdot|s) = \mathrm{softmax}\{Q^{\text{ITER}}(s, \cdot)/\beta\}, \forall s$.

---

This condition fails only in degenerate multi-objective settings when for *every* state $s \in \mathcal{S}$, the set $S_{\text{center}}(s)$ lies entirely within an affine subspace of dimension less than $K + L$. Under this assumption, the Hessian $H[\mathcal{L}(u, w)]$ is positive definite. (See Appendix F.1 for more details.) Let $\lambda$ denote the minimum eigenvalue of $H[\mathcal{L}(u, w)]$, which satisfies $0 < \lambda \le \alpha$ (Bubeck, 2015). Theorem 3.6 provides a formal guarantee of convergence for Algorithm 1 under approximate Q-updates.

**Theorem 3.6.** *Let $(u^*, w^*)$ denote an optimal solution to equation 7. For each outer-loop index $m \ge 1$ in Algorithm 1, let $Q^*_{u^m, w^m}$ denote the fixed point of equation 13 with $[u; w] = [u^m; w^m]$, and let $Q^m$ denote the Q-function after completing the $m$-th inner-loop update. For each $m$, assume $\|Q^m - Q^*_{u^m, w^m}\|_\infty < \epsilon$ for some $\epsilon > 0$. Then for $m \ge 1$,*

$$\|[u^m; w^m] - [u^*; w^*]\|_2 \le (1 - \frac{\lambda}{\alpha})^m \|[u^0; w^0] - [u^*; w^*]\|_2 + \frac{\sqrt{|\mathcal{S}|}}{\lambda}\sqrt{\sum_{i=1}^{K+L}\{r_{max}^{(i)}\}^2}\frac{1 + \gamma}{(1 - \gamma)^2}\epsilon. \quad (15)$$

*(Proof: See Appendix F.2.)*

Theorem 3.6 establishes that the error decreases geometrically at rate $O\left((1 - \frac{\lambda}{\alpha})^m\right)$, up to $O(\epsilon)$. (For completeness, Appendix F.3 provides the analysis of the degenerate case without *Assumption*.)

### 3.3 DISCUSSION

| | $w$ fixed | $w$ learned |
|---|---|---|
| $L = 0$ | Unconstr. weight-sum (Yang et al., 2019) | Unconstr. max-min (Park et al., 2024) |
| $L \ge 1$ | Constr. weight-sum (Huang et al., 2021) | Constr. max-min |

Table 1: Generalizability of our framework to previous MORL settings

Our new framework is general enough to unify many existing MORL formulations. Note that we have two major design choices: (i) scalarization strategy: whether the preference vector $w$ on $K$ objectives used in the scalarization function is fixed or learned/adaptive, and (ii) whether constraints are present. Table 1 shows four different setups of our framework. Our framework covers unconstrained weight-sum MORL with $L = 0$ and fixed $w$, constrained weighted-sum MORL with $L \ge 1$ and fixed $w$, unconstrained max-min MORL with $L = 0$ and $w$ learning, and finally constrained max-min MORL with $L \ge 1$ and $w$ learning.

## 4 RELATED WORK

**MORL** The dominant approach in MORL is utility-based (Roijers et al., 2013; Hayes et al., 2022), where the goal is to find an optimal policy $\pi^* = \arg\max_\pi f(J(\pi))$ given a non-decreasing

scalarization function $f : \mathbb{R}^K \to \mathbb{R}$. When $f$ is linear, each non-negative weight vector defines a scalarized MDP (Boutilier et al., 1999), motivating work on learning a single model capable of generating policies across a continuum of preferences (Abels et al., 2019; Yang et al., 2019; Basaklar et al., 2023; Hung et al., 2023; Lu et al., 2023; Park & Sung, 2025; Li et al., 2025). This family of approaches is known as multi-policy MORL (Roijers et al., 2013; Hayes et al., 2022). For non-linear scalarization functions, however, Bellman optimality no longer holds in its standard form due to the loss of linearity, making optimization substantially more difficult (Roijers et al., 2013; Hayes et al., 2022). Algorithms that directly optimize the scalarized objective belong to the single-policy MORL category (Roijers et al., 2013; Hayes et al., 2022). Most work in this category considers a welfare function (Siddique et al., 2020; Cousins et al., 2024) as the nonlinear scalarization $f$. Note that single-policy and multi-policy methods are complementary rather than interchangeable (Roijers et al., 2013; Hayes et al., 2022).

**Unconstrained Max-min MORL**   Max-min MORL studies the case where $f = \min$, aiming to enforce max-min fairness. This formulation is useful in many applications such as mitigating bottlenecks in cloud and edge resource management systems (Saifullah et al., 2014; Wang et al., 2019). Several studies optimize proxy objectives related to the unconstrained max-min formulation, for example, maximizing a conservative lower bound of $\mathbb{E}_\pi \left[ \min_{1 \le k \le K} \left( \sum_{t=0}^\infty \gamma^t r_t^{(k)} \right) \right]$ (Fan et al., 2023; Peng et al., 2025), or maximizing the total return while enforcing per-group performance constraints (Eaton et al., 2025).

The work most closely related to ours is Park et al. (2024), which proposes a tractable approach for exact unconstrained max-min MORL using Gaussian smoothing to estimate gradients. However, this approach requires maintaining multiple network copies, increasing computational overhead. Furthermore, the gradient estimates are inherently inexact, as Gaussian smoothing of a convex function yields a convex upper bound rather than the true function (Nesterov & Spokoiny, 2017). In contrast, our method produces direct, theoretically grounded gradient estimates and extends naturally to constrained MORL. Concurrent with our work, Byeon et al. (2025) introduced an alternative unconstrained max-min MORL formulation based on a two-player zero-sum game framework (Daskalakis & Panageas, 2018; Miryoosefi et al., 2019). However, their method does not address the incorporation of constraints.

**Constrained RL**   Many approaches to constrained MDPs reformulate the problem with a scalar reward (i.e., a special case of equation 2 and equation 3 with $K = 1$ and without $f$) into an unconstrained one by augmenting the objective with a weighted sum of constraint violations, typically via a Lagrangian formulation (Achiam et al., 2017; Tessler et al., 2018; Paternain et al., 2019; Ha et al., 2020; Vaswani et al., 2022; Calvo-Fullana et al., 2023; Müller et al., 2024). The motivation for this line of work is that the Lagrangian relaxation exhibits no duality gap, even when the original problem is non-convex with respect to the policy (Paternain et al., 2019). Most methods in this category, therefore, rely on alternating updates between the policy and the Lagrange multipliers. However, these approaches do not consider the multi-objective reward setting in equation 2 and equation 3 with $K \ge 2$. Moreover, applying them directly to our setting is non-trivial, since $f = \min$ introduces non-differentiability in equation 2.

To resolve this, we reformulate our problem as a convex program using occupancy measures and then derive another convex program equivalent to the dual problem, which serves as the basis for our MORL algorithm. In particular, we show that both the max-min criterion and the constraints can be satisfied by jointly updating the weights $u$ and $w$, a simple yet effective approach that to our knowledge has not been explored in the constrained MDP literature. Although Lee et al. (2022) also leverages convex analysis with occupancy measures, its focus is on constrained single-objective RL with a scalar reward (i.e., $K = 1$) in an offline setting. Unlike our work, it does not address fairness across multiple objectives in MORL settings.

Several recent works have incorporated constraints into MORL (Huang et al., 2021; Lin et al., 2024; Kim et al., 2025; Liu et al., 2025), but under settings different from our framework which integrates max-min optimization. See Appendix G for details of these works.

# 5 EXPERIMENTS

In this section, we present experimental validations of our theoretical analysis and algorithm. Section 5.1 examines the convergence properties of our method in tabular settings. In Section 5.2, we further demonstrate the practical relevance of our approach through applications including edge computing resource allocation and multi-objective locomotion control.

## 5.1 TABULAR SETTINGS

We conducted experiments in tabular settings to evaluate the convergence of our algorithm. Constrained MOMDPs were randomly generated, after a feasibility check, within two widely used classes of structured MDPs. (See Appendix H.1 for details on the feasibility check.) First, bipartite state graphs partition the state space into two disjoint subsets, enforcing transitions between them at alternating time steps. This structure captures temporal dynamics in systems with role alternation or interleaving phases (Littman, 1994). Second, hierarchical MDPs organize the state space into multiple levels or stages, where transitions flow sequentially from one level to the next. This reflects tasks with subgoals or temporal abstraction (Dieterich, 2000).

The optimal value for each MOMDP was computed by solving equation 4, equation 5, and equation 6 with $\beta = 0$ via linear programming (LP), and performance was evaluated as the error relative to these LP-optimal values. We compared our method, which computes $\nabla_u v_{u,w}^*(s)$ and $\nabla_w v_{u,w}^*(s)$ using Theorem 3.3, against a modified version of the Gaussian smoothing method from Park et al. (2024). We adapted this baseline to incorporate both the max-min weights ($w$) and the constraint weights ($u$). We selected this baseline because, to the best of our knowledge, no prior work has proposed a constrained max-min MORL algorithm. However, Park et al. (2024) can be naturally extended to this setting using its gradient estimation approach based on Gaussian smoothing. Both methods follow the same alternating update scheme: (i) updating the policy using equation 13 and (ii) updating the weight vectors using projected gradient descent, until convergence with respect to $(u, w)$. (See Appendix H.2 for further details on the baseline and experimental setup.)

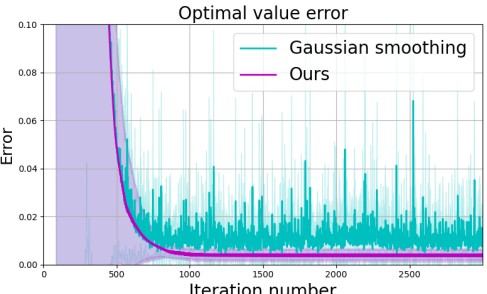

| Algorithm | Optimal value error ($\downarrow$) |
|---|---|
| Ours | 0.004 |
| w/o $u$ update | 0.325 |
| w/o $w$ update | 0.657 |
| w/o $(u, w)$ upd. | 1.008 |

Figure 1: Average optimality gap across generated MOMDPs

Table 2: Ablation study on the impact of weight learning in our tabular setting

Figure 1 shows that our method converges reliably to the optimal value, whereas the Gaussian smoothing baseline exhibits larger approximation errors and unstable learning behavior. This observation can be explained by two factors: (i) Gaussian smoothing of a convex function yields another convex function that forms an upper bound to the original objective, and (ii) its theoretical guarantee ensures only average-iterate convergence rather than last-iterate convergence (Nesterov & Spokoiny, 2017). The latter contributes to the oscillatory behavior observed during training.

As analyzed in Appendix H.3, the Gaussian smoothing baseline requires approximately $N + 1$ times more computation per weight update than our method, where $N$ is the number of perturbed Q-tables used in Gaussian smoothing. In summary, our method is superior in accuracy and computation for constrained max-min optimization compared to Gaussian smoothing in tabular settings.

To evaluate the effect of learning the weight vectors in our algorithm, we independently disabled the learning of $u$, of $w$, and of both $(u, w)$ while initializing $u$ to a zero vector and $w = [1/K, \cdots, 1/K] \in \Delta^K$ on the simplex. Table 2 demonstrates that removing the learning

of either weight component noticeably increases the optimal value estimation error. (See Appendix H.4 for the ablation study on the impact of the regularization coefficient $\beta$.)

## 5.2 Extension to Applications

In this section, we extend our algorithm to practical applications. To ensure stable gradient estimation of our algorithm in continuous state spaces, we parameterize a gradient network $g_\theta(s) \in \mathbb{R}^{L+K}$ to estimate $\nabla_u v_{u,w}^*(s)$ and $\nabla_w v_{u,w}^*(s)$, following Theorem 3.3. Implementation details, including gradient estimation and our constrained max-min algorithm for applications, are provided in Appendix I.1.

### 5.2.1 Edge Computing Resource Allocation

We consider a simulated edge computing resource allocation environment (Bae et al., 2020). The system includes $N_{\text{type}}$ distinct user application types, and multiple mobile devices generate tasks according to these types and send them to an edge computing node. The edge computing node is equipped with multi-core CPUs and maintains $N_{\text{type}}$ separate task queues, each associated with a specific application type. Incoming tasks from the mobile devices are sorted into these queues accordingly. Once tasks arrive, the edge computing node either processes them locally or offloads a portion to a cloud computing node through a dedicated communication link.

The unconstrained reward is an $N_{\text{type}}$-dimensional vector, where each entry corresponds to the negative value of the current queue length for a given application type to encourage queue minimization. Minimizing the delay of the worst-performing user group is crucial for maintaining smooth system operation (Zehavi et al., 2013; Saifullah et al., 2014; Wang et al., 2019). The cost is the total power consumption of the system, normalized by the environment. The goal is to control the system to minimize the maximum cumulative discounted sum of queue length across application types within each episode, while satisfying the system's power consumption constraint with its designed threshold value $C_{th} = 5.6$ with $N_{\text{type}} = 3$. (Additional details of the environment are given in Appendix I.2.)

We consider five baselines: (i) randomly selects one queue for allocation at each timestep (Random), (ii) unconstrained max-average SAC (MA-SAC) (Haarnoja et al., 2018), (iii) max-average SAC with a Lagrangian relaxation (MA-SAC-L) (Ha et al., 2020; Yang et al., 2021), (iv) unconstrained max-min MORL algorithm with Gaussian smoothing Park et al. (2024) (Max-min GS), and (v) unconstrained max-min MORL algorithm from a concurrent work (Byeon et al., 2025) (ARAM). Each of the baselines lacks either max-min fairness ((iii)), constraint handling ((iv),(v)), or both ((i), (ii)). We report the mean performance computed across twelve random seeds. (See Appendices I.3 and I.4 for the implementation of the Max-min baseline GS and hyperparameter settings, respectively.)

| Algorithm | Cost sum ($C_{th} = 5.6$) | Maximum queue length ($\downarrow$) |
|---|---|---|
| Random | 5.9 | 72.4 |
| MA-SAC | 5.8 | 46.5 |
| MA-SAC-L | **5.6** | 52.9 |
| Ours | **5.6** | 37.9 |
| Max-min GS | 5.8 | 23.7 |
| ARAM | 6.1 | 14.8 |

Table 3: Cumulative cost sum and total maximum queue length with the two constraint-satisfying algorithms highlighted in **bold**

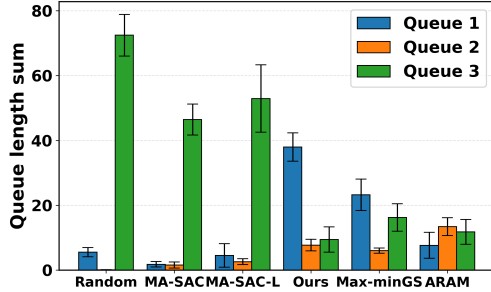

Figure 2: Comparison of queue length sums across queues for each algorithm

Table 3 presents the cumulative cost sum and the total maximum queue length. Compared to the Random baseline, MA-SAC reduces the total maximum queue length but still fails to satisfy the power consumption constraint, with its cost sum exceeding the threshold $C_{th} = 5.6$. While MA-SAC-L satisfies the power constraint, it does so at the cost of a higher total maximum queue length compared to MA-SAC. As shown in Figure 2, our method substantially reduces the total maximum queue length

relative to MA-SAC-L, while still adhering to the power constraint. We note that both Max-min GS and ARAM violate the power constraint.

Table 4 shows that ablating the constraint-related $u$ update causes constraint violations, while removing the max-min-related $w$ update substantially increases the total maximum queue length. These results confirm that our method effectively balances max-min performance with constraint satisfaction.

| Algorithm | Cost sum $(C_{th} = 5.6)$ | Maximum queue length ($\downarrow$) |
|---|---|---|
| Ours | 5.6 | 37.9 |
| w/o $u$ update | 5.8 | 33.7 |
| w/o $w$ update | 5.5 | 52.7 |
| w/o $(u, w)$ upd. | 5.8 | 44.7 |

Table 4: Ablation study in resource allocation

### 5.2.2 MULTI-OBJECTIVE LOCOMOTION CONTROL

We include MoAnt-v5 environment (Felten et al., 2023), where the agent learns locomotion to maximize $x$ and $y$ velocities while keeping energy consumption under a threshold. We consider an asymmetric case where movement in the $x$ direction is attenuated by friction at rate 0.3. The velocities $(v_x, v_y)$, combined with bonus terms, constitute a 2-D reward, while the control cost is treated as a constraint. (See Appendix I.5 for details on hyperparameters.)

| Algorithm | Cost sum $(C_{th} = 50)$ | Minimum return ($\uparrow$) |
|---|---|---|
| Random | 146.5 | 48.2 |
| MA-SAC | 275.3 | 98.8 |
| MA-SAC-L | **47.8** | 83.0 |
| Ours | **28.3** | 92.2 |
| Max-min GS | 111.7 | 92.7 |
| ARAM | 620.7 | 101.3 |

Table 5: MoAnt-v5 results over five seeds with the two constraint-satisfying algorithms highlighted in **bold**

Table 5 shows that both our method and MA-SAC-L satisfy the constraints, but our method achieves superior max-min performance. In contrast, the other four algorithms severely violate the constraints, as they do not explicitly account for constraint satisfaction. Overall, our algorithm balances constraint satisfaction and max-min fairness.

In Appendix J, we further evaluate our method on a traffic signal control environment (Alegre, 2019) featuring 16 objectives (Park & Sung, 2025; Byeon et al., 2025), allowing us to test its performance in a higher-dimensional MORL setting.

## 6 CONCLUSION

We have proposed a unified framework for constrained MORL that integrates max-min fairness with constraint satisfaction. Our approach offers flexibility in modeling problems that satisfy fairness and operational constraints. We established a theoretical foundation and developed an algorithm that shows strong performance in both tabular settings and practical applications. By jointly addressing fairness and resource constraints, our work contributes to advancing sustainable AI, offering a compelling alternative to conventional approaches that focus solely on performance, often at the expense of equity and resource constraints. A broader impact of our work is discussed in Appendix K, and a discussion of limitations and future directions is provided in Appendix L.

### REPRODUCIBILITY STATEMENT

We provide detailed descriptions of our algorithm in Section 3.2 and Appendix I.1. Appendices H and I contain the experimental setup, fine-tuned hyperparameters, and infrastructure details. To ensure accessibility and reproducibility, we provide the source code for the resource allocation environment in the supplementary material. Furthermore, all theorems are presented in a self-contained manner, making it straightforward to verify the theoretical results.

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

# A    PROOF ON OPTIMALITY GAP

*Proof.* With a slight abuse of notation, let $J(\pi) := [J_1(\pi), \cdots, J_K(\pi)]^\top \in \mathbb{R}^K$ and let $\mathcal{H}(\pi)$ denote the expected cumulative entropy of $\pi$. We express the optimization of equation 2 and equation 3 with $f = \min$ as follows:

$$\max_{\pi \in \Pi_{\text{feas}}} \min_{1 \leq k \leq K} J_k(\pi) + \beta \mathcal{H}(\pi) \tag{16}$$

where $\Pi_{\text{feas}} := \left\{ \pi \mid \mathbb{E}_{\mu_0, \pi} \left[ \sum_{t=0}^\infty \gamma^t c_t^{(l)} \right] \geq C^{(l)}, \quad \forall l = 1, \ldots, L \right\}$ and it is assumed to be non-empty under the typical assumption in constrained RL (Tessler et al., 2018; Ha et al., 2020).

Let the optimal solution to the regularized problem in equation 16 be $\pi_r^* := \arg\max_{\pi \in \Pi_{\text{feas}}} \min_{1 \leq k \leq K} J_k(\pi) + \beta \mathcal{H}(\pi) = \arg\max_{\pi \in \Pi_{\text{feas}}} \min_w \langle w, J(\pi) \rangle + \beta \mathcal{H}(\pi)$ where $\min_{w \in \Delta^K}$ is abbreviated as $\min_w$ for brevity. Let $w^*(\pi) := \arg\min_w \langle w, J(\pi) \rangle$ and $w_r^* := w^*(\pi_r^*)$.

Let the optimal solution to the unregularized problem be $\pi^* := \arg\max_{\pi \in \Pi_{\text{feas}}} \min_w \langle w, J(\pi) \rangle$ and $w^* = w^*(\pi^*)$. Let the optimal max-min value of the unregularized problem be $V_{w^*}^{\pi^*} := \langle w^*, J(\pi^*) \rangle$. Similarly, let the optimal value of the regularized problem be $V_{w_r^*}^{\pi_r^*} := \langle w_r^*, J(\pi_r^*) \rangle$. For simplicity, we abbreviate $\max_{\pi \in \Pi_{\text{feas}}}$ as $\max_\pi$ below.

First, a lower bound is derived as follows:

$V_{w_r^*}^{\pi_r^*} + \beta \mathcal{H}(\pi_r^*)$

$= \max_\pi \min_w \langle w, J(\pi) \rangle + \beta \mathcal{H}(\pi)$

$\geq \min_w \langle w, J(\pi^*) \rangle + \beta \mathcal{H}(\pi^*)$

$= \langle w^*, J(\pi^*) \rangle + \beta \mathcal{H}(\pi^*).$

Since $0 \leq \mathcal{H}(\pi) \leq \frac{\log |\mathcal{A}|}{1-\gamma}$ for any $\pi$, we obtain $V_{w_r^*}^{\pi_r^*} - V_{w^*}^{\pi^*} \geq -\frac{\beta \log |\mathcal{A}|}{1-\gamma}$.

Next, an upper bound is derived as follows:

$V_{w^*}^{\pi^*}$

$= \max_\pi \min_w \langle w, J(\pi) \rangle$

$\geq \min_w \langle w, J(\pi_r^*) \rangle$

$= \langle w_r^*, J(\pi_r^*) \rangle.$

Thus, $V_{w_r^*}^{\pi_r^*} - V_{w^*}^{\pi^*} \leq 0$.

Combining these two bounds, we obtain the optimality value gap ranges as $0 \leq V_{w^*}^{\pi^*} - V_{w_r^*}^{\pi_r^*} \leq \frac{\beta \log |\mathcal{A}|}{1-\gamma}$.  $\square$

# B  PROOF ON EQUIVALENT OPTIMIZATION

*Proof.* The dual problem of equation 4, equation 5, and equation 6 is rewritten as follows:

$$\min_{u \geq 0} \min_{w \geq 0, v} \min_{\xi \geq 0} \max_{\rho, b} \left[ b(1 - \sum_{k=1}^{K} w_k) - \beta \sum_{s,a} \rho(s,a) \log \frac{\rho(s,a)}{\sum_{a'} \rho(s,a')} \right.$$

$$+ \sum_{s} \mu_0(s) v(s) + \sum_{s,a} \xi(s,a) \rho(s,a) - \sum_{l=1}^{L} u_l C^{(l)}$$

$$\left. + \sum_{s,a} \rho(s,a) [\sum_{k=1}^{K} w_k r^{(k)}(s,a) + \sum_{l=1}^{L} u_l c^{(l)}(s,a) + \gamma \sum_{s'} T(s'|s,a) v(s') - v(s)] \right]. \tag{17}$$

Here $b$ is an auxiliary variable satisfying $\sum_{s,a} r^{(k)}(s,a) \rho(s,a) \geq b$, $1 \leq k \leq K$. Let $\eta_{u,v,w}(s,a) := \sum_{k=1}^{K} w_k r^{(k)}(s,a) + \sum_{l=1}^{L} u_l c^{(l)}(s,a) + \gamma \sum_{s'} T(s'|s,a) v(s') - v(s)$. We apply KKT conditions.

1. Stationarity condition gives

$$\forall (s,a), \quad -\beta \log \frac{\rho(s,a)}{\sum_{a'} \rho(s,a')} + \xi(s,a) + \eta_{u,v,w}(s,a) = 0 \tag{18}$$

and

$$1 - \sum_{k=1}^{K} w_k = 0. \tag{19}$$

2. Complementary slackness condition gives
$$\forall (s,a), \quad \xi(s,a) \rho(s,a) = 0. \tag{20}$$

From equation 18, we derive

$$\forall (s,a), \quad \frac{\rho(s,a)}{\sum_{a'} \rho(s,a')} = \exp\left( \frac{\xi(s,a) + \eta_{u,v,w}(s,a)}{\beta} \right) \tag{21}$$

so $\rho(s,a) > 0$ and $\xi(s,a) = 0$ from equation 20. Therefore,

$$\forall (s,a), \quad \frac{\rho(s,a)}{\sum_{a'} \rho(s,a')} = \exp\left( \frac{\eta_{u,v,w}(s,a)}{\beta} \right). \tag{22}$$

Inserting equation 19 and equation 22, we obtain:

$$\min_{u \in \mathbb{R}_+^L} \min_{v,w} \sum_{s} \mu_0(s) v(s) - \sum_{l=1}^{L} u_l C^{(l)} \tag{23}$$

$$\forall s, \quad v(s) = \beta \log \sum_{a} \exp[\frac{1}{\beta} \{ \sum_{k=1}^{K} w_k r^{(k)}(s,a) + \sum_{l=1}^{L} u_l c^{(l)}(s,a) + \gamma \sum_{s'} T(s'|s,a) v(s') \}] := [\mathcal{T}_{u,w} v](s)$$
$$\tag{24}$$

$$\sum_{k=1}^{K} w_k = 1; \; w_k \geq 0 \; \forall 1 \leq k \leq K. \tag{25}$$

where equation 24 is derived from $\sum_{a} \exp\left( \frac{\eta_{u,v,w}(s,a)}{\beta} \right) = 1$, $\forall s$, and strong duality holds under Slater condition (Boyd & Vandenberghe, 2004). Since $\mathcal{T}_{u,w}$ is a contraction mapping (Haarnoja et al., 2017; Fox et al., 2016), it has the unique fixed point $v^*_{u,w}$. Therefore, $v = v^*_{u,w}$ is the only feasible solution that satisfies equation 24 and we have the following:

$$\min_{u \in \mathbb{R}_+^L, w \in \Delta^K} \mathcal{L}(u,w) = \sum_{s} \mu_0(s) v^*_{u,w}(s) - \sum_{l=1}^{L} u_l C^{(l)}. \tag{26}$$

Under Slater condition, this optimization attains the same optimal value as in the original convex optimization. Lastly, the convexity of this optimization is directly obtained from Theorem 4.1. in Park et al. (2024).

□

## C    Proof of Differentiability

*Proof.* We first note that for the simplicity of notation, it is enough to show the theorem for the case of $L = 0$ (i.e., with no constraints). This holds because, given $(u, w) \in \mathbb{R}^{L+K}$, the mapping $\mathcal{T}_{u,w}$ is defined by $[\mathcal{T}_{u,w}v](s) = \beta \log \sum_a \exp[\frac{1}{\beta}\{\sum_{l=1}^{L} u_l c^{(l)}(s, a) + \sum_{k=1}^{K} w_k r^{(k)}(s, a) + \gamma \sum_{s'} T(s'|s, a)v(s')\}], \forall s$, and we can regard the concatenation of $c(s, a)$ and $r(s, a)$ as a new vector reward of size $L + K$ with its weight $(u, w)$. Therefore, we use the notation of the following mapping $[\mathcal{T}_w v](s) = \beta \log \sum_a \exp[\frac{1}{\beta}\{\sum_{k=1}^{K} w_k r^{(k)}(s, a) + \gamma \sum_{s'} T(s'|s, a)v(s')\}], \forall s$.

Let $|\mathcal{S}| = p$. We define $F(w, v) := v - \mathcal{T}_w v$, $F : \mathbb{R}^K \times \mathbb{R}^p \to \mathbb{R}^p$. Let $v_w^*$ be the unique fixed point of $\mathcal{T}_w$. Then $F(w, v_w^*) = v_w^* - \mathcal{T}_w v_w^* = 0$. Here $v_w^*$ is implicitly expressed w.r.t. $w$, and we aim to analyze $v_w^*$ using *implicit function theorem*.

First of all, $F : \mathbb{R}^K \times \mathbb{R}^p \to \mathbb{R}^p$ is a continuously differentiable function. For each $s$, $F(w, v)(s) = v(s) - [\mathcal{T}_w v](s) = v(s) - \beta \log \sum_a \exp[\frac{1}{\beta}\{\sum_{k=1}^{K} w_k r^{(k)}(s, a) + \gamma \sum_{s'} T(s'|s, a)v(s')\}]$ which is a composition of linear, logarithm, summation, exponential, and linear functions.

Now we fix $w$ and check whether the Jacobian matrix $\partial_v F(w, v)|_{v=v_w^*} \in \mathbb{R}^{p \times p}$ is invertible where $[\partial_v F(w, v)|_{v=v_w^*}]_{ij} = \frac{\partial F(w,v)(s_i)}{\partial v(s_j)}|_{v=v_w^*}$. We have $\partial_v F(w, v) = I_p - \partial_v[\mathcal{T}_w v]$ where $I_p$ is the $p \times p$ identity matrix. Then

$$\frac{\partial [\mathcal{T}_w v](s_i)}{\partial v(s_j)}\Big|_{v=v_w^*} = \gamma \mathbb{E}_{a \sim \pi_w^*(\cdot|s_i)}[T(s_j|s_i, a)] \tag{27}$$

where

$$\pi_w^*(a|s) = \frac{\exp[\frac{1}{\beta}\{\sum_{k=1}^{K} w_k r^{(k)}(s, a) + \gamma \sum_{s'} T(s'|s, a)v_w^*(s')\}]}{\sum_{a'} \exp[\frac{1}{\beta}\{\sum_{k=1}^{K} w_k r^{(k)}(s, a') + \gamma \sum_{s'} T(s'|s, a')v_w^*(s')\}]}. \tag{28}$$

If we denote $T(\cdot|s, a) := [T(s_1|s, a) \cdots T(s_p|s, a)]$, we have

$$\partial_v F(w, v)|_{v=v_w^*} = I_p - \gamma \begin{bmatrix} \mathbb{E}_{a \sim \pi_w^*(\cdot|s_1)}[T(\cdot|s_1, a)] \\ \vdots \\ \mathbb{E}_{a \sim \pi_w^*(\cdot|s_p)}[T(\cdot|s_p, a)] \end{bmatrix} =: I_p - \gamma \begin{bmatrix} T^{\pi_w^*}(\cdot|s_1) \\ \vdots \\ T^{\pi_w^*}(\cdot|s_p) \end{bmatrix} \tag{29}$$

where $T^{\pi_w^*}(s_j|s_i) = \mathbb{E}_{a \sim \pi_w^*(\cdot|s_i)}[T(s_j|s_i, a)] =: [T^{\pi_w^*}]_{ij}$. Then $I_p - \gamma T^{\pi_w^*}$ is invertible since $T^{\pi_w^*}$ is a row-stochastic square matrix (Horn & Johnson, 2012).

Therefore, $\partial_v F(w, v)|_{v=v_w^*}$ is invertible. By implicit function theorem, there exists an open set $U \subset \mathbb{R}^K$ containing $w$ such that there exists a unique continuously differentiable function $h : U \to \mathbb{R}^p$ such that $h(w) = v_w^*$ and $F(w', h(w')) = 0$, i.e., $h(w') = \mathcal{T}_{w'} h(w')$ for all $w' \in U$. Since $h(w')$ is the unique fixed point of $\mathcal{T}_{w'}$, $h(w') = v_{w'}^*, \forall w' \in U$. If we use the implicit function theorem for all $w \in \mathbb{R}^K$, we can conclude that $v = v_w^*$ is a unique continuously differentiable function in $w \in \mathbb{R}^K$ that satisfies $v = \mathcal{T}_w v$.

Moreover, for $1 \leq k \leq K$,

$$\frac{\partial [\mathcal{T}_w v](s_i)}{\partial w_k}\Big|_{v=v_w^*} = \mathbb{E}_{a \sim \pi_w^*(\cdot|s_i)}[r^{(k)}(s_i, a)]. \tag{30}$$

With a slight abuse of notation, if we denote $r(s, a) := [r^{(1)}(s, a) \cdots r^{(K)}(s, a)]$, we have

$$\partial_w F(w, v)|_{v=v_w^*} = - \begin{bmatrix} \mathbb{E}_{a \sim \pi_w^*(\cdot|s_1)}[r(s_1, a)] \\ \vdots \\ \mathbb{E}_{a \sim \pi_w^*(\cdot|s_p)}[r(s_p, a)] \end{bmatrix} =: - \begin{bmatrix} r^{\pi_w^*}(s_1) \\ \vdots \\ r^{\pi_w^*}(s_p) \end{bmatrix} \tag{31}$$

where $r^{\pi_w^*}(s) = \mathbb{E}_{a \sim \pi_w^*(\cdot|s)}[r(s, a)] \in \mathbb{R}^{1 \times K}$. By implicit function theorem, we have

$$\begin{bmatrix} \nabla_w v_w^*(s_1)^\top \\ \vdots \\ \nabla_w v_w^*(s_p)^\top \end{bmatrix} = -[\partial_v F(w, v)|_{v=v_w^*}]^{-1} \partial_w F(w, v)|_{v=v_w^*} = (I_p - \gamma T^{\pi_w^*})^{-1} r^{\pi_w^*}. \tag{32}$$

Note that the $k$-th ($1 \le k \le K$) column of equation 32 is equivalent to the policy evaluation of $\pi_w^*$ considering a scalar reward function $r^{(k)}$ (Silver, 2015; Sutton & Barto, 2018). We denote the value function as $v_k^{\pi_w^*} \in \mathbb{R}^p$. Then

$$\frac{\partial v_w^*(s)}{\partial w_k} = v_k^{\pi_w^*}(s), \ \forall s. \tag{33}$$

If we denote $v^{\pi_w^*}(s) = [v_1^{\pi_w^*}(s), \cdots, v_K^{\pi_w^*}(s)]^\top \in \mathbb{R}^K$ for all $s$, then $v^{\pi_w^*}(s)$ is the value function evaluated with the policy $\pi_w^*$ in a given MOMDP. We have

$$\nabla_w v_w^*(s) = v^{\pi_w^*}(s), \ \forall s. \tag{34}$$

For the case of $L > 0$, the only difference is that $\pi_w^*$ is changed to

$$\pi_{u,w}^*(a|s) = \frac{\exp[\frac{1}{\beta}\{\sum_{l=1}^{L} u_l c^{(l)}(s,a) + \sum_{k=1}^{K} w_k r^{(k)}(s,a) + \gamma\sum_{s'}T(s'|s,a)v_{u,w}^*(s')\}]}{\sum_{a'} \exp[\frac{1}{\beta}\{\sum_{l=1}^{L} u_l c^{(l)}(s,a') + \sum_{k=1}^{K} w_k r^{(k)}(s,a') + \gamma\sum_{s'}T(s'|s,a')v_{u,w}^*(s')\}]} \tag{35}$$

where $v_{u,w}^*$ is the fixed point of the operator $\mathcal{T}_{u,w}$:

$$\forall s, \ [\mathcal{T}_{u,w}v](s) = \beta \log \sum_a \exp[\frac{1}{\beta}\{\sum_{l=1}^{L} u_l c^{(l)}(s,a) + \sum_{k=1}^{K} w_k r^{(k)}(s,a) + \gamma\sum_{s'}T(s'|s,a)v(s')\}] \tag{36}$$

and the column size of $r^{\pi_{u,w}^*}$ is $L + K$, not $K$. We denote $v_c^{\pi_{u,w}^*}(s) \in \mathbb{R}^L$, $v_r^{\pi_{u,w}^*}(s) \in \mathbb{R}^K$ as the value functions evaluated with the policy $\pi_{u,w}^*$ for constrained reward $c$ and unconstrained reward $r$, respectively. Finally, we have

$$\nabla_u v_{u,w}^*(s) = v_c^{\pi_{u,w}^*}(s), \ \nabla_w v_{u,w}^*(s) = v_r^{\pi_{u,w}^*}(s), \ \forall s. \tag{37}$$

$\square$

## D  PROOF OF TWICE-DIFFERENTIABILITY

*Proof.* Here we also use the implicit function theorem and follow a similar logic in the proof of differentiability in Appendix C. Let $|\mathcal{S}| = p$. We show the theorem for the case of $L = 0$ to guarantee notational simplicity. For each $1 \le i \le K$, we want to show that $\frac{\partial v_w^*}{\partial w_i} := [\frac{\partial v_w^*(s_1)}{\partial w_i}, \cdots, \frac{\partial v_w^*(s_p)}{\partial w_i}]^\top \in \mathbb{R}^p$ is differentiable in $w \in \mathbb{R}^K$. From the result in Appendix C, we have

$$\frac{\partial v_w^*}{\partial w_i} = v_i^{\pi_w^*} \tag{38}$$

where $v_i^{\pi_w^*} \in \mathbb{R}^p$ is the value function evaluated with the policy $\pi_w^*$ in equation 28 with the $i$-th reward $r^{(i)}$. Let $r_i^{\pi_w^*}(s) = \mathbb{E}_{a \sim \pi_w^*(\cdot|s)}[r^{(i)}(s, a)] \in \mathbb{R}$. From equation 32, we have

$$v_i^{\pi_w^*} = (I_p - \gamma T^{\pi_w^*})^{-1} r_i^{\pi_w^*} \tag{39}$$

or equivalently,

$$v_i^{\pi_w^*} = r_i^{\pi_w^*} + \gamma T^{\pi_w^*} v_i^{\pi_w^*} =: \mathcal{T}_w^* v_i^{\pi_w^*}. \tag{40}$$

We define $F(w, v) := v - \mathcal{T}_w^* v$, $F : \mathbb{R}^K \times \mathbb{R}^p \to \mathbb{R}^p$. Then $F(w, v_i^{\pi_w^*}) = v_i^{\pi_w^*} - \mathcal{T}_w v_i^{\pi_w^*} = 0$. Here $v_i^{\pi_w^*}$ is the unique fixed point of $\mathcal{T}_w^*$ and is implicitly expressed w.r.t. $w$, and we aim to analyze $v_i^{\pi_w^*}$ using *implicit function theorem*.

First of all, $F : \mathbb{R}^K \times \mathbb{R}^p \to \mathbb{R}^p$ is a continuously differentiable function. For each $s$, $F(w, v)(s) = v(s) - [\mathcal{T}_w^* v](s) = v(s) - [r_i^{\pi_w^*}(s) + \gamma \sum_{s'} T^{\pi_w^*}(s'|s) v(s')] = v(s) - \sum_a \pi_w^*(a|s)[r^{(i)}(s, a) + \gamma \sum_{s'} T(s'|s, a) v(s')]$. As seen in equation 28, $\pi_w^*$ contains $v_w^*$ which is continuously differentiable in $w$ (as a result of the proof in Appendix C), and $\pi_w^*$ is a composition of quotient, exponential, summation and linear functions of $w$ and $v_w^*$.

Now we fix $w$ and check whether the Jacobian matrix $\partial_v F(w, v)|_{v=v_i^{\pi_w^*}} \in \mathbb{R}^{p \times p}$ is invertible where $[\partial_v F(w, v)|_{v=v_i^{\pi_w^*}}]_{ij} = \frac{\partial F(w,v)(s_i)}{\partial v(s_j)}|_{v=v_i^{\pi_w^*}}$. We have $\partial_v F(w, v) = I_p - \partial_v[\mathcal{T}_w^* v]$ where $I_p$ is the $p \times p$ identity matrix. Then

$$\frac{\partial[\mathcal{T}_w^* v](s_i)}{\partial v(s_j)}\Big|_{v=v_i^{\pi_w^*}} = \gamma \mathbb{E}_{a \sim \pi_w^*(\cdot|s_i)}[T(s_j|s_i, a)]. \tag{41}$$

If we denote $T(\cdot|s, a) := [T(s_1|s, a) \cdots T(s_p|s, a)]$, we have

$$\partial_v F(w, v)|_{v=v_i^{\pi_w^*}} = I_p - \gamma \begin{bmatrix} \mathbb{E}_{a \sim \pi_w^*(\cdot|s_1)}[T(\cdot|s_1, a)] \\ \vdots \\ \mathbb{E}_{a \sim \pi_w^*(\cdot|s_p)}[T(\cdot|s_p, a)] \end{bmatrix} =: I_p - \gamma \begin{bmatrix} T^{\pi_w^*}(\cdot|s_1) \\ \vdots \\ T^{\pi_w^*}(\cdot|s_p) \end{bmatrix} \tag{42}$$

where $T^{\pi_w^*}(s_j|s_i) = \mathbb{E}_{a \sim \pi_w^*(\cdot|s_i)}[T(s_j|s_i, a)] =: [T^{\pi_w^*}]_{ij}$. Then $I_p - \gamma T^{\pi_w^*}$ is invertible since $T^{\pi_w^*}$ is a row-stochastic square matrix (Horn & Johnson, 2012).

Therefore, $\partial_v F(w, v)|_{v=v_i^{\pi_w^*}}$ is invertible. By implicit function theorem, there exists an open set $U \subset \mathbb{R}^K$ containing $w$ such that there exists a unique continuously differentiable function $h : U \to \mathbb{R}^p$ such that $h(w) = v_i^{\pi_w^*}$ and $F(w', h(w')) = 0$, i.e., $h(w') = \mathcal{T}_{w'}^* h(w')$ for all $w' \in U$. Since $h(w')$ is the unique fixed point of $\mathcal{T}_{w'}^*$, $h(w') = v_i^{\pi_{w'}^*}, \forall w' \in U$. If we use the implicit function theorem for all $w \in \mathbb{R}^K$, we can conclude that $v = v_i^{\pi_w^*}$ is a unique continuously differentiable function in $w \in \mathbb{R}^K$ that satisfies $v = \mathcal{T}_w^* v$.

Now, for $1 \le j \le K$, we aim to calculate $\frac{\partial[\mathcal{T}_w^* v](s)}{\partial w_j}|_{v=v_i^{\pi_w^*}}$. For notational simplicity, let $Q_w^*(s, a) := \sum_{k=1}^K w_k r^{(k)}(s, a) + \gamma \sum_{s'} T(s'|s, a) v_w^*(s')$. Then we express $\pi_w^*$ as follows:

$$\pi_w^*(a|s) = \frac{\exp[\frac{1}{\beta}\{Q_w^*(s, a)\}]}{\sum_{a'} \exp[\frac{1}{\beta}\{Q_w^*(s, a')\}]}. \tag{43}$$

We also have

$$\frac{\partial Q_w^*(s,a)}{\partial w_j} = r^{(j)}(s,a) + \gamma\sum_{s'}T(s'|s,a)\frac{\partial v_w^*(s')}{\partial w_j} = r^{(j)}(s,a) + \gamma\sum_{s'}T(s'|s,a)v_j^{\pi_w^*}(s') := Q_j^{\pi_w^*}(s,a).$$

(44)

In other words, we denote $Q_j^{\pi_w^*}$ as the action-value function evaluated with $\pi_w^*$ for a scalar reward function $r^{(j)}$. Then

$$\frac{\partial[\mathcal{T}_w^* v](s)}{\partial w_j}\Big|_{v=v_i^{\pi_w^*}} = \sum_a Q_i^{\pi_w^*}(s,a)\frac{\partial \pi_w^*(a|s)}{\partial w_j}$$

(45)

which is equivalent to

$$\frac{\partial[\mathcal{T}_w^* v](s)}{\partial w_j}\Big|_{v=v_i^{\pi_w^*}} = \frac{1}{\beta}\sum_a Q_i^{\pi_w^*}(s,a)\left[\pi_w^*(a|s)Q_j^{\pi_w^*}(s,a) - \pi_w^*(a|s)\sum_{a'}\{\pi_w^*(a'|s)Q_j^{\pi_w^*}(s,a')\}\right]$$

(46)

and we have

$$\frac{\partial[\mathcal{T}_w^* v](s)}{\partial w_j}\Big|_{v=v_i^{\pi_w^*}} = \frac{1}{\beta}\left[\mathbb{E}_{a\sim\pi_w^*(\cdot|s)}[Q_i^{\pi_w^*}(s,a)Q_j^{\pi_w^*}(s,a)] - \mathbb{E}_{a\sim\pi_w^*(\cdot|s)}[Q_i^{\pi_w^*}(s,a)]\mathbb{E}_{a\sim\pi_w^*(\cdot|s)}[Q_j^{\pi_w^*}(s,a)]\right].$$

(47)

By implicit function theorem, we have

$$\begin{bmatrix} \nabla_w \frac{\partial v_w^*(s_1)}{\partial w_i}^\top \\ \vdots \\ \nabla_w \frac{\partial v_w^*(s_p)}{\partial w_i}^\top \end{bmatrix} = -[\partial_v F(w,v)|_{v=v_i^{\pi_w^*}}]^{-1}\partial_w F(w,v)|_{v=v_i^{\pi_w^*}} = \frac{1}{\beta}(I_p - \gamma T^{\pi_w^*})^{-1}E_i^{\pi_w^*} \quad (48)$$

where $E_i^{\pi_w^*}$ is a $p \times K$ matrix where for each row corresponding to $s$, the $j$-th element is $\mathbb{E}_{a\sim\pi_w^*(\cdot|s)}[Q_i^{\pi_w^*}(s,a)Q_j^{\pi_w^*}(s,a)] - \mathbb{E}_{a\sim\pi_w^*(\cdot|s)}[Q_i^{\pi_w^*}(s,a)]\mathbb{E}_{a\sim\pi_w^*(\cdot|s)}[Q_j^{\pi_w^*}(s,a)]\}$. This formulation holds for each $1 \le i \le K$.

Therefore, we construct a $p \times K \times K$ tensor, say $B^{\pi_w^*}$, by stacking $\{E_i^{\pi_w^*}\}_i$ along the new (third) dimension. Then along the first dimension of size $p$, for each $s$, let $B^{\pi_w^*}(s) \in \mathbb{R}^{K\times K}$ be the corresponding slice of $B$. Let $Q^{\pi_w^*}(s,a) = [Q_1^{\pi_w^*}(s,a), \cdots, Q_K^{\pi_w^*}(s,a)]^\top \in \mathbb{R}^K$ be the action-value function evaluated with $\pi_w^*$ for vector reward $r$. Then we have

$$B^{\pi_w^*}(s) = \mathbb{E}_{a\sim\pi_w^*(\cdot|s)}\left[(Q^{\pi_w^*}(s,a) - \mathbb{E}_{a'\sim\pi_w^*(\cdot|s)}[Q^{\pi_w^*}(s,a')])(Q^{\pi_w^*}(s,a) - \mathbb{E}_{a'\sim\pi_w^*(\cdot|s)}[Q^{\pi_w^*}(s,a')])^\top\right]$$

(49)

which is the covariance matrix of $Q^{\pi_w^*}(s,\cdot)$ over the probability distribution $\pi_w^*(\cdot|s)$. Let $s_k$ correspond to the $k$-th row of $T^{\pi_w^*}$ ($1 \le k \le p$). Then we have the following Hessian formulation for $s_k$:

$$H[v_w^*(s_k)] = \frac{1}{\beta}\sum_{l=1}^p [(I_p - \gamma T^{\pi_w^*})^{-1}]_{kl}B^{\pi_w^*}(s_l).$$

(50)

For the case of $L > 0$, the only difference is that $\pi_w^*$ is changed to

$$\pi_{u,w}^*(a|s) = \frac{\exp[\frac{1}{\beta}\{\sum_{l=1}^L u_l c^{(l)}(s,a) + \sum_{k=1}^K w_k r^{(k)}(s,a) + \gamma\sum_{s'}T(s'|s,a)v_{u,w}^*(s')\}]}{\sum_{a'}\exp[\frac{1}{\beta}\{\sum_{l=1}^L u_l c^{(l)}(s,a') + \sum_{k=1}^K w_k r^{(k)}(s,a') + \gamma\sum_{s'}T(s'|s,a')v_{u,w}^*(s')\}]}$$

(51)

where $v_{u,w}^*$ is the fixed point of the operator $\mathcal{T}_{u,w}$:

$$\forall s, [\mathcal{T}_{u,w}v](s) = \beta\log\sum_a\exp[\frac{1}{\beta}\{\sum_{l=1}^L u_l c^{(l)}(s,a) + \sum_{k=1}^K w_k r^{(k)}(s,a) + \gamma\sum_{s'}T(s'|s,a)v(s')\}] \quad (52)$$

and the size of $B^{\pi^*_{u,w}}(s)$ is $(L+K) \times (L+K)$, not $K \times K$, defined by $Q^{\pi^*_{u,w}}(s,a) \in \mathbb{R}^{L+K}$ which is the action-value function evaluated with $\pi^*_{u,w}$ for the concatenated vector function of constrained reward $c$ and unconstrained reward $r$. Finally, we have

$$H[v^*_{u,w}(s_k)] = \frac{1}{\beta} \sum_{l=1}^{p} [(I_p - \gamma T^{\pi^*_{u,w}})^{-1}]_{kl} B^{\pi^*_{u,w}}(s_l). \tag{53}$$

$\square$

## E    PROOF OF SMOOTHNESS

*Proof.* Let $a = (u', w')$ and $b = (u'', w'')$ in $\mathbb{R}^{L+K}$. By the differentiability of $\nabla v^*_{u,w}(s)$ proved in Theorem 3.4, we use generalized mean value inequality in Banach spaces and have

$$\|\nabla v^*_{u,w}(s)|_{(u,w)=b} - \nabla v^*_{u,w}(s)|_{(u,w)=a}\|_2 \leq \sup_{t \in [0,1]} \|H[v^*_{u,w}(s)]|_{(u,w)=a+t(b-a)}\|_2 \|b - a\|_2 \tag{54}$$

Let $\lambda_{\max}(A)$ be the maximum eigenvalue of a real symmetric matrix $A$. For each $s_k$ ($1 \leq k \leq p$), the eigenvalues of $H[v^*_{u,w}(s_k)]$ are nonnegative. Since trace operator is additive, we have

$$\|H[v^*_{u,w}(s_k)]\|_2 = \lambda_{\max}(H[v^*_{u,w}(s_k)]) \leq \mathrm{Tr}(H[v^*_{u,w}(s_k)]) = \frac{1}{\beta} \sum_{l=1}^{p} [(I_p - \gamma T^{\pi^*_{u,w}})^{-1}]_{kl} \mathrm{Tr}(B^{\pi^*_{u,w}}(s_l)). \tag{55}$$

For each $s$, we also have

$$\mathrm{Tr}(B^{\pi^*_{u,w}}(s)) = \sum_{k=1}^{L+K} \mathrm{Var}(Q_k^{\pi^*_{u,w}}(s,a)) \leq \sum_{k=1}^{L+K} \mathbb{E}[|Q_k^{\pi^*_{u,w}}(s,\cdot)|^2] \leq \sum_{k=1}^{L+K} \left( \frac{r_{\max}^{(k)}}{1-\gamma} \right)^2. \tag{56}$$

Since $(I_p - \gamma T^{\pi^*_{u,w}})^{-1} = \sum_{i=0}^{\infty}(\gamma T^{\pi^*_{u,w}})^i$ and each $(T^{\pi^*_{u,w}})^i$ is a probability transition matrix,

$$\|H[v^*_{u,w}(s_k)]\|_2 \leq \frac{1}{\beta} \sum_{m=1}^{L+K} \left( \frac{r_{\max}^{(m)}}{1-\gamma} \right)^2 \left( \sum_{i=0}^{\infty} \gamma^i \sum_{l=1}^{p} (T^{\pi^*_{u,w}})_{kl}^i \right) = \frac{1}{\beta(1-\gamma)} \sum_{m=1}^{L+K} \left( \frac{r_{\max}^{(m)}}{1-\gamma} \right)^2. \tag{57}$$

It should be noted that $\|H[v^*_{u,w}(s_k)]\|_2$ is uniformly bounded regardless of $s_k$ and $(u,w)$. Therefore, $\nabla v^*_{u,w}(s)$ is Lipschitz continuous in $\|\cdot\|_2$ from equation 54. $\square$

## F CONVERGENCE ANALYSIS

### F.1 ASSUMPTION FOR ACTION-VALUE NONDEGENERACY

*Assumption* There exists at least one state $s \in \mathcal{S}$ such that the centered action-value vectors $\left\{ Q^{\pi^*_{u,w}}(s,a) - \mathbb{E}_{a' \sim \pi^*_{u,w}(\cdot|s)}[Q^{\pi^*_{u,w}}(s,a')] : a \in \mathcal{A} \right\}$ span $\mathbb{R}^{K+L}$.

This condition fails only in degenerate multi-objective settings when for *every* state $s \in \mathcal{S}$, the set $\left\{ Q^{\pi^*_{u,w}}(s,a) - \mathbb{E}_{a' \sim \pi^*_{u,w}(\cdot|s)}[Q^{\pi^*_{u,w}}(s,a')] : a \in \mathcal{A} \right\}$ lies entirely within an affine subspace of dimension less than $K + L$ (e.g., the size of an action set is smaller than the number of objectives).

Then $B^{\pi^*_{u,w}}(s) = \mathbb{E}_{a \sim \pi^*_{u,w}(\cdot|s)}\left[ (Q^{\pi^*_{u,w}}(s,a) - \mathbb{E}_{a' \sim \pi^*_{u,w}(\cdot|s)}[Q^{\pi^*_{u,w}}(s,a')])(Q^{\pi^*_{u,w}}(s,a) - \mathbb{E}_{a' \sim \pi^*_{u,w}(\cdot|s)}[Q^{\pi^*_{u,w}}(s,a')])^\top \right] \in \mathbb{R}^{(L+K) \times (L+K)}$ is positive definite. This is because (i) $\pi^*_{u,w}(a|s) > 0$ for all $a$ (equation 10, which has this favorable property that facilitate analysis), and (ii) for any $y \in \mathbb{R}^{K+L}$ with $y \neq \mathbf{0}$, $y^\top B^{\pi^*_{u,w}}(s)y = \sum_a \pi^*_{u,w}(a|s)\left( y^\top (Q^{\pi^*_{u,w}}(s,a) - \mathbb{E}_{a' \sim \pi^*_{u,w}(\cdot|s)}[Q^{\pi^*_{u,w}}(s,a')]) \right)^2 > 0$ as at least one $a$ should satisfy $y^\top (Q^{\pi^*_{u,w}}(s,a) - \mathbb{E}_{a' \sim \pi^*_{u,w}(\cdot|s)}[Q^{\pi^*_{u,w}}(s,a')]) \neq 0$.

By Theorem 3.4, we have the Hessian of $\mathcal{L}(u,w)$ as $H[\mathcal{L}(u,w)] = \frac{1}{\beta}\sum_{l=1}^p [\mu_0^\top (I_p - \gamma T^{\pi^*_{u,w}})^{-1}]_l B^{\pi^*_{u,w}}(s_l) = \frac{1}{\beta}\sum_s \rho^{\pi^*_{u,w}}(s)B^{\pi^*_{u,w}}(s)$ where $p = |\mathcal{S}|$ and $\rho^{\pi^*_{u,w}}(s) = \sum_{t=0}^\infty \gamma^t \mathrm{Pr}(s_t = s|\pi^*_{u,w}, \mu_0)$, and $\rho^{\pi^*_{u,w}}(s) > 0$ by the reachability assumption (Lee et al., 2021). Therefore, $H[\mathcal{L}(u,w)]$ is positive definite under the assumption.

### F.2 PROOF OF CONVERGENCE ANALYSIS

Let $\lambda_{\min}(A)$ be the minimum eigenvalue of a real symmetric matrix $A$. For simplicity, we denote $\lambda := \lambda_{\min}(H[\mathcal{L}(u,w)])$. Then $0 < \lambda \leq \alpha$ (Bubeck, 2015) and $\mathcal{L}(u,w)$ is $\lambda$-strongly convex.

**Theorem 3.6** Let $(u^*, w^*)$ denote the optimal solution to equation 7. For each outer-loop index $m \geq 1$ in Algorithm 1, let $Q^*_{u^m, w^m}$ denote the fixed point of equation 13 with $[u;w] = [u^m; w^m]$, and let $Q^m$ denote the Q-function after completing the $m$-th inner-loop update. For each $m$, assume $\|Q^m - Q^*_{u^m, w^m}\|_\infty < \epsilon$ for some $\epsilon > 0$. Then for $m \geq 1$,

$$\|[u^m; w^m] - [u^*; w^*]\|_2 \leq (1 - \frac{\lambda}{\alpha})^m \|[u^0; w^0] - [u^*; w^*]\|_2 + \frac{\sqrt{|\mathcal{S}|}}{\lambda}\sqrt{\sum_{i=1}^{K+L}\{r_{\max}^{(i)}\}^2}\frac{1+\gamma}{(1-\gamma)^2}\epsilon. \quad (58)$$

*Proof.* By the definition in equation 10, we have the optimal policy $\pi^*_{u^m, w^m}(a|s) = \frac{\exp(\frac{1}{\beta}Q^*_{u^m, w^m}(s,a))}{\sum_{a'}\exp(\frac{1}{\beta}Q^*_{u^m, w^m}(s,a'))}$ when $(u,w) = (u^m, w^m)$. According to Theorem 3.3, we have $\nabla_{(u,w)}\mathcal{L}(u^m, v^m) = [\sum_s \mu_0(s)v_c^{\pi^*_{u^m, v^m}}(s) - [C^{(1)}, \cdots, C^{(L)}]^\top; \sum_s \mu_0(s)v_r^{\pi^*_{u^m, v^m}}(s)] \in \mathbb{R}^{L+K}$. We also have $\tilde{\nabla}_{(u,w)}\mathcal{L}(u^m, v^m) := [\sum_s \mu_0(s)v_c^{\pi^m}(s) - [C^{(1)}, \cdots, C^{(L)}]^\top; \sum_s \mu_0(s)v_r^{\pi^m}(s)] \in \mathbb{R}^{L+K}$, an estimated gradient of $\nabla_{(u,w)}\mathcal{L}(u^m, w^m)$ using $\pi^m$ where $\pi^m(a|s) = \frac{\exp(\frac{1}{\beta}Q^m(s,a))}{\sum_{a'}\exp(\frac{1}{\beta}Q^m(s,a'))}$.

Let $e_m := \tilde{\nabla}_{(u,w)}\mathcal{L}(u^m, v^m) - \nabla_{(u,w)}\mathcal{L}(u^m, w^m)$. For each $s$, let $v_{r,i}^\pi(s)$ ($1 \leq i \leq K$) and $v_{c,j}^\pi(s)$ ($1 \leq j \leq L$) denote the elements of the $i$-th dimension of $v_r^\pi(s) \in \mathbb{R}^K$ and the $j$-th

dimension of $v_c^\pi(s) \in \mathbb{R}^L$, respectively. Then we have

$$\|e_m\|_2^2 = \|[\sum_s \mu_0(s)(v_c^{\pi^m}(s) - v_c^{\pi_{u^m,v^m}^*}(s)); \sum_s \mu_0(s)(v_r^{\pi^m}(s) - v_r^{\pi_{u^m,v^m}^*}(s))]\|_2^2$$

$$= \sum_{i=1}^{K}\left(\sum_s \mu_0(s)(v_{r,i}^{\pi^m}(s) - v_{r,i}^{\pi_{u^m,w^m}^*}(s))\right)^2 + \sum_{j=1}^{L}\left(\sum_s \mu_0(s)(v_{c,j}^{\pi^m}(s) - v_{c,j}^{\pi_{u^m,w^m}^*}(s))\right)^2$$

$$\leq \|\mu_0\|_2^2 \sum_s\left[\sum_{i=1}^{K}(v_{r,i}^{\pi^m}(s) - v_{r,i}^{\pi_{u^m,w^m}^*}(s))^2 + \sum_{j=1}^{L}(v_{c,j}^{\pi^m}(s) - v_{c,j}^{\pi_{u^m,w^m}^*}(s))^2\right] \quad (59)$$

where $\|\mu_0\|_2^2 = \sum_s(\mu_0(s))^2$ and the inequality holds by Cauchy-Schwarz.

Since both $\pi^m$ and $\pi_{u^m,w^m}^*$ use softmax parameterization with $Q^m$ and $Q_{u^m,w^m}^*$, respectively, we have

$$\forall s, \ |v_{r,i}^{\pi^m}(s) - v_{r,i}^{\pi_{u^m,w^m}^*}(s)| \leq \frac{(1+\gamma)r_{\max}^{(i)}}{(1-\gamma)^2}\|Q^m - Q_{u^m,w^m}^*\|_\infty \ (1 \leq i \leq K) \quad (60)$$

and

$$\forall s, \ |v_{c,j}^{\pi^m}(s) - v_{c,j}^{\pi_{u^m,w^m}^*}(s)| \leq \frac{(1+\gamma)r_{\max}^{(K+j)}}{(1-\gamma)^2}\|Q^m - Q_{u^m,w^m}^*\|_\infty \ (1 \leq j \leq L) \quad (61)$$

according to the property of equation (261) in Yang et al. (2024). Combining equation 60, equation 61, and $\|\mu_0\|_2 \leq 1$ with equation 59 gives

$$\|e_m\|_2 \leq \sqrt{|\mathcal{S}|}\sqrt{\sum_{i=1}^{K+L}\{r_{\max}^{(i)}\}^2}\frac{1+\gamma}{(1-\gamma)^2}\|Q^m - Q_{u^m,w^m}^*\|_\infty$$

$$< \sqrt{|\mathcal{S}|}\sqrt{\sum_{i=1}^{K+L}\{r_{\max}^{(i)}\}^2}\frac{1+\gamma}{(1-\gamma)^2}\epsilon. \quad (62)$$

Next, we view the projected gradient descent for each outer loop as a proximal gradient descent. We reformulate the optimization in equation 7 of

$$\min_{u \in \mathbb{R}_+^L, w \in \Delta^K} \mathcal{L}(u, w) \quad (63)$$

as follows:

$$\min_{(u,w) \in \mathbb{R}^{L+K}} \mathcal{L}(u, w) + I_{\mathbb{R}_+^L \times \Delta^K}(u, w) \quad (64)$$

where $I_{\mathbb{R}_+^L \times \Delta^K}(u, w)$ is the indicator function with its value 0 if $(u, w) \in \mathbb{R}_+^L \times \Delta^K$ and $+\infty$ otherwise. $I_{\mathbb{R}_+^L \times \Delta^K}$ is convex because its epigraph $\{(u, w, t_e)|t_e \geq 0, (u, w) \in \mathbb{R}_+^L \times \Delta^K\}$ is convex. We note that according to Theorem 3.5, the smoothness of $\mathcal{L}(u, w)$ is satisfied on $\mathbb{R}^{L+K}$, which makes equation 64 valid. We also note that we computed the smoothness coefficient $\alpha = \frac{1}{\beta(1-\gamma)}\sum_{i=1}^{K+L}\left(\frac{r_{\max}^{(i)}}{1-\gamma}\right)^2$ of $\mathcal{L}$ in Appendix E.

Applying the error bound in equation 62 to the analysis of inexact proximal gradient method (Schmidt et al., 2011), we have

$$\|[u^m; w^m] - [u^*; w^*]\|_2 \leq (1 - \frac{\lambda}{\alpha})^m\|[u^0; w^0] - [u^*; w^*]\|_2 + \frac{1}{\alpha}\sum_{i=1}^{m}(1 - \frac{\lambda}{\alpha})^{m-i}\|e_i\|_2$$

$$\leq (1 - \frac{\lambda}{\alpha})^m\|[u^0; w^0] - [u^*; w^*]\|_2 + \frac{\sqrt{|\mathcal{S}|}}{\lambda}\sqrt{\sum_{i=1}^{K+L}\{r_{\max}^{(i)}\}^2}\frac{1+\gamma}{(1-\gamma)^2}\epsilon.$$

$$(65)$$

This is achieved because we use the convex optimization method from Wang & Carreira-Perpiñán (2013) for projection onto the simplex $\Delta^K$, and apply non-negativity clipping for projection onto $\mathbb{R}_+^L$,

both of them induce zero error in each phase of proximal objective update as it is fully deterministic and avoids randomized procedures.

It remains to check whether $I_{\mathbb{R}_+^L \times \Delta^K}$ in equation 64 is a lower semi-continuous proper convex function (Schmidt et al., 2011). $I_{\mathbb{R}_+^L \times \Delta^K}$ is lower semi-continuous because $\mathbb{R}_+^L \times \Delta^K$ is closed, and it is also proper convex since $I_{\mathbb{R}_+^L \times \Delta^K}$ never attains $-\infty$ and $\mathbb{R}_+^L \times \Delta^K$ is non-empty.

$\square$

### F.3 CONVERGENCE ANALYSIS FOR DEGENERATE CASE

**Theorem F.1.** *Let $(u^*, w^*)$ denote an optimal solution to equation 7. For each outer-loop index $m \geq 1$ in Algorithm 1, let $Q^*_{u^m, w^m}$ denote the fixed point of equation 13 with $[u; w] = [u^m; w^m]$, and let $Q^m$ denote the Q-function after completing the $m$-th inner-loop update. For each $m$, assume $\|Q^m - Q^*_{u^m, w^m}\|_\infty < \epsilon_m$ for some $\epsilon_m > 0$. Then for $m \geq 1$,*

$$\mathcal{L}(\frac{1}{m}\sum_{i=1}^m (u^i, w^i)) - \mathcal{L}(u^*, w^*) \leq \frac{\alpha}{2m}(\|[u^0; w^0] - [u^*; w^*]\|_2 \; + \frac{2M}{\alpha}\sum_{i=1}^m \epsilon_i)^2 \tag{66}$$

*where $M = \sqrt{|\mathcal{S}|}\sqrt{\sum_{j=1}^{K+L}\{r_{max}^{(j)}\}^2}\frac{1+\gamma}{(1-\gamma)^2}$.*

*Proof.* Using an analysis of inexact proximal gradient method (Schmidt et al., 2011) using the same logic in the proof of Theorem 3.6 (Appendix F.2), we have

$$\mathcal{L}(\frac{1}{m}\sum_{i=1}^m (u^i, w^i)) - \mathcal{L}(u^*, w^*) \leq \frac{\alpha}{2m}(\|[u^0; w^0] - [u^*; w^*]\|_2 \; + \frac{2}{\alpha}\sum_{i=1}^m \|e_i\|_2)^2 \tag{67}$$

where $e_i := \tilde{\nabla}_{(u,w)}\mathcal{L}(u^i, w^i) - \nabla_{(u,w)}\mathcal{L}(u^i, w^i)$ is the $i$-th gradient error and

$$\|e_i\|_2 < \sqrt{|\mathcal{S}|}\sqrt{\sum_{j=1}^{K+L}\{r_{max}^{(j)}\}^2}\frac{1+\gamma}{(1-\gamma)^2}\epsilon_i = M\epsilon_i \tag{68}$$

from equation 62. $\square$

We note that the error of $\mathcal{L}(\frac{1}{m}\sum_{i=1}^m (u^i, w^i)) - \mathcal{L}(u^*, w^*)$ decreases at rate $O(\frac{1}{m})$ when $\{\epsilon_i\}_{i=1}^\infty$ is summable (e.g., $\epsilon_m = O(\frac{1}{m^{1+\delta}})$ with $\delta > 0$).

## G  ADDITIONAL RELATED WORK

Several recent works have explored constrained MORL, but under settings that differ from ours, which explicitly incorporates the max-min objective.

For instance, Huang et al. (2021) reformulates constrained RL as a MOMDP by treating constraint costs as an additional reward dimension, thereby enabling constraint satisfaction while exploring preference trade-offs. Similarly, Kim et al. (2025) learns preference-conditioned policies by reformulating the agent update to mitigate objective-wise gradient conflicts. These approaches pursue complementary goals to ours. Both are based on a multi-policy framework, where policies are conditioned on a preference vector. However, it remains unclear how to select a preference vector such that the resulting policy exactly corresponds to the optimal solution under a nonlinear scalarization function, such as the max-min criterion considered in our setting. In contrast, our method directly solves a single-policy constrained max-min optimization problem. This conceptual distinction parallels the complementary relationship between single-policy and multi-policy approaches in unconstrained MORL (Roijers et al., 2013; Hayes et al., 2022).

Recently, Lin et al. (2024) studied offline constrained MORL, where policies are trained on offline data and later adapted to target preferences using additional demonstrations. In contrast, our work focuses on online learning and does not assume access to additional demonstration data. Liu et al. (2025) train multiple policies in parallel to approximate the Pareto front, improving coverage by solving constrained optimizations in underexplored regions. Their method targets standard MORL with linear scalarization, enhancing it via constrained optimization rather than directly tackling constrained MORL.

## H  EXPERIMENTAL DETAILS: TABULAR SETTINGS

### H.1  FEASIBILITY CHECK

When generating structured MOMDPs randomly, we first verify whether the generated instances are feasible. To do this, We first consider the following unregularized convex optimization:

$$\max_{\rho \geq 0} \min_{1 \leq k \leq K} \left( \sum_{(s,a)} r^{(k)}(s,a)\rho(s,a) \right) \tag{69}$$

$$\sum_{a'} \rho(s',a') = \mu_0(s') + \gamma \sum_{(s,a)} T(s'|s,a)\rho(s,a), \ \forall s' \tag{70}$$

$$\sum_{(s,a)} c^{(l)}(s,a)\rho(s,a) \geq C^{(l)}, \ l = 1, \cdots, L \tag{71}$$

which is equivalently expressed as the following LP by using additional scalar variable $\tilde{c} \in \mathbb{R}$:

$$\max_{\rho \geq 0, \tilde{c}} \tilde{c} \tag{72}$$

$$\sum_{a'} \rho(s',a') = \mu_0(s') + \gamma \sum_{(s,a)} T(s'|s,a)\rho(s,a), \ \forall s' \tag{73}$$

$$\sum_{(s,a)} r^{(k)}(s,a)\rho(s,a) \geq \tilde{c}, \ k = 1, \cdots, K, \tag{74}$$

$$\sum_{(s,a)} c^{(l)}(s,a)\rho(s,a) \geq C^{(l)}, \ l = 1, \cdots, L. \tag{75}$$

We want to generate $\mu_0, T, r$, and $c$ in structured MOMDPs to satisfy feasibility and Slater condition by solving the following LP using the pywraplp function from the OR-Tools library:

$$\max_{\rho \geq \epsilon_{\text{low}}} 0 \tag{76}$$

$$\sum_{a'} \rho(s', a') = \mu_0(s') + \gamma \sum_{(s,a)} T(s'|s, a)\rho(s, a), \ \forall s' \tag{77}$$

$$\sum_{(s,a)} r^{(k)}(s, a)\rho(s, a) \geq \tilde{c} + \epsilon_{\text{low}}, \ \ k = 1, \cdots, K, \tag{78}$$

$$\sum_{(s,a)} c^{(l)}(s, a)\rho(s, a) \geq C^{(l)} + \epsilon_{\text{low}}, \ \ l = 1, \cdots, L \tag{79}$$

where $\epsilon_{\text{low}}$ is used to guarantee the strict feasibility for Slater condition, and we set $\epsilon_{\text{low}} = 10^{-4}$. If the LP solver does not find a feasible solution, we regenerate the constrained MOMDP until a feasible instance is found. Once any feasible solution is found, we solve the LP of equation 72, equation 73, equation 74, and equation 75 by using LP solver to acquire the optimal max-min value $\tilde{c}^*$.

## H.2 EXPERIMENTAL SETUP

In the Gaussian smoothing method, we create $N$ copies $\{\tilde{Q}_i\}_{i=1}^N$ of the current $Q$-function and update each $\tilde{Q}_i$ using scalarization with $N$ perturbed weights $\{(\tilde{u}_i, \tilde{w}_i)\}_{i=1}^N$, sampled from a Gaussian distribution centered at the current weight vector $(u, w)$. Specifically, we compute $\tilde{Q}_i(s, a) \leftarrow [\tilde{u}_i; \tilde{w}_i]^\top [c; r] + \gamma \sum_{s'} T(s'|s, a)\beta \log \sum_{a'} \exp\left(\frac{\tilde{Q}_i(s', a')}{\beta}\right)$ until convergence, given the perturbed weights $\{(\tilde{u}_i, \tilde{w}_i)\}_{i=1}^N$. The gradient w.r.t. $(u, w)$ is then estimated by computing the slope of a linear regression over the pairs $[\{(\tilde{u}_i, \tilde{w}_i)\}_{i=1}^N, \{\tilde{Q}_i\}_{i=1}^N]$.

The update of our algorithm is applied iteratively for each $(u, w)$ pair until the maximum change in the $Q$-function between successive iterations falls below $10^{-4}$. We use the following setting: $\gamma = 0.8$, $l_{\text{w}} = 0.001$, and ITER $= 3000$. $u$ was initialized as all-one vector while $w$ is initialized as the uniform vector on the simplex. For Gaussian smoothing, we set $N = 24$ and use a Gaussian distribution with a standard deviation 0.001. We tuned $N$ to prevent unstable divergence in the Gaussian smoothing method when $N$ is too small, while also avoiding excessive computational overhead. Both algorithms used $\beta = 0.03$ for the bipartite setting and $\beta = 0.01$ for the hierarchical setting, respectively. Each algorithm was evaluated using three random seeds for each constrained MOMDP setting, resulting in six runs when averaged across the MOMDP classes. All experiments were conducted on an Intel Core i9-10900X CPU @ 3.70GHz.

## H.3 COMPARISON OF ALGORITHMIC COMPLEXITY

We now include a comparison of the algorithmic complexity per weight update $(u, w)$ in tabular settings. Let $S = |\mathcal{S}|$, $A = |\mathcal{A}|$, and $d = K + L$. Although each update of equation 13 given weight $(u, w)$ theoretically requires infinitely many steps for convergence, we denote the practical number of steps as $T_{\text{soft}}$ for our complexity analysis.

First, the per-iteration complexity of our method is given by $O(T_{\text{soft}}S^2A + SAd + S^3 + S^2d)$. Here, $T_{\text{soft}}S^2A$ is the cost of update in equation 13, and the remaining part is the cost of computing the gradient via dynamic programming based on Theorem 3.3. If $T_{\text{soft}}$ is large enough, the update of equation 13 dominates the computation: $O(T_{\text{soft}}S^2A + SAd + S^3 + S^2d) \approx O(T_{\text{soft}}S^2A)$.

Regarding the Gaussian smoothing method, let $N$ denote the number of perturbed Q-tables used for smoothing. Then the complexity per iteration is $O((N + 1)T_{\text{soft}}S^2A + d^3 + Nd^2)$ where $(N+1)T_{\text{soft}}S^2A$ is the computation of equation 13 for the current Q-table and its $N$ copies. The other terms are related to gradient estimation using linear regression (Park et al., 2024). Again, equation 13 dominates the computation and $O((N + 1)T_{\text{soft}}S^2A + d^3 + Nd^2) \approx O((N + 1)T_{\text{soft}}S^2A)$ if $T_{\text{soft}}$ is large enough.

In summary, the Gaussian smoothing baseline incurs approximately $N + 1$ times more computational cost per weight update compared to our method. Note that the complexity of the projection onto $\Delta^K$ is $O(K \log K)$ (Wang & Carreira-Perpiñán, 2013) which is relatively lightweight compared to other components, due to the sublinear growth of the logarithmic term.

## H.4 IMPACT OF THE ENTROPY REGULARIZATION COEFFICIENT

We further analyzed the effect of $\beta$ on convergence. As shown in Table 6, values of $\beta < 0.1$ yield stable convergence with low sensitivity across both constrained MOMDP types. Based on this, we recommend selecting $\beta$ such that the ratio between (i) the entropy term (scaled by $\beta$) and (ii) the objective value without entropy remains below $10^{-2}$, approximating the optimal value as $\beta \to 0$.

| $\beta$ | 0.1 | 0.03 | 0.01 | 0.003 | 0.001 |
|---|---|---|---|---|---|
| Bipartite | 0.086 | 0.005 | 0.016 | 0.029 | 0.026 |
| Hierarchical | 0.037 | 0.004 | 0.003 | 0.011 | 0.016 |

Table 6: Optimal value errors of our algorithm across different values of $\beta$

# I  EXPERIMENTAL DETAILS: APPLICATIONS

## I.1  IMPLEMENTATION OF OUR ALGORITHM FOR APPLICATIONS

We now leverage the usage of neural network for our algorithm. If we differentiate the both side of $v_{u,w}^*(s) = [\mathcal{T}_{u,w}v_{u,w}^*](s)$ w.r.t. $u$ and $w$ for all $s$, then we have the following formula:

$$\forall s, \ \nabla_w v_{u,w}^*(s) = \sum_a \pi_{u,w}^*(a|s)\left(r(s,a) + \gamma\sum_{s'}T(s'|s,a)\nabla_w v_{u,w}^*(s')\right). \tag{80}$$

$$\forall s, \ \nabla_u v_{u,w}^*(s) = \sum_a \pi_{u,w}^*(a|s)\left(c(s,a) + \gamma\sum_{s'}T(s'|s,a)\nabla_u v_{u,w}^*(s')\right). \tag{81}$$

Here, $\pi_{u,w}^*(a|s)$ is defined as in equation 10. To ensure stable gradient estimation in continuous state spaces, we parameterize a gradient network to estimate $\nabla_u v_{u,w}^*(s)$ and $\nabla_w v_{u,w}^*(s)$. Since each action is also continuous, we employ an actor network $\pi_\theta$ and implement Algorithm 2. To further stabilize the estimation of the gradient, we add an additional linear layer after the penultimate layer of the actor network $\pi_\theta$, and use its $(L + K)$-dimensional output as the gradient network $g_\theta(s)$. We use the notation $g_\theta$ to indicate that the actor network and the gradient network share parameters and jointly update their lower-layer weights.

---

**Algorithm 2** Proposed Constrained Max-min Algorithm for Continuous Action

---

1: $\pi_\theta$: actor, $Q_\phi$: critic, $Q_{\overline{\phi}}$: target critic, $g_\theta$: gradient network, $g_{\overline{\theta}}$: target gradient network, $\mathcal{D}$: replay buffer, $T_{\text{init}}$: initial iteration number, $\tau$: target update ratio, $U$: main iteration number, $U_s$: gradient step for critic update, $l_g$: learning rate of the gradient network, $l_0$: initial learning rate of the weight $(u, w)$, $K$: unconstrained reward dimension, $L$: the number of constraints, $C_{th} \in \mathbb{R}^L$: threshold vector for the constraints

2: Initialize target critic $\overline{\phi} \leftarrow \phi$, target gradient network $\overline{\theta} \leftarrow \theta$, and weights $u^0 \in \mathbb{R}_+^L$, $w^0 \in \Delta^K$.

3: **for** $j = 0, \cdots, T_{\text{init}} - 1$ **do**

4:     Rollout sample from $\pi_\theta$ and save it in $\mathcal{D}$. Sample a batch of data $\mathcal{B} \subset \mathcal{D}$.

5:     $Q_\phi \leftarrow$ **Critic Update**$(Q_\phi, Q_{\overline{\phi}}, \pi_\theta, (u^0, w^0), \mathcal{B})$ (Algorithm 3)

6:     Update target critic parameter $\overline{\phi} \leftarrow \tau\phi + (1 - \tau)\overline{\phi}$.

7:     $\pi_\theta \leftarrow$ **Actor Update**$(Q_\phi, \pi_\theta, \mathcal{D})$ (Algorithm 4)

8: **end for**

9: **for** $m = 0, 1, 2, \cdots, U - 1$ **do**

10:     Rollout sample from $\pi_\theta$ and save $(s, a, r, c, s', \pi_{\theta_{\text{old}}}(a|s))$ in $\mathcal{D}$ where $\pi_{\theta_{\text{old}}}(a|s) = \pi_\theta(a|s)$.

11:     Update the gradient network $g_\theta$ as follows:

$$\theta \leftarrow \theta - l_g \nabla_\theta \mathbb{E}_{(s,a,r,c,s',\pi_{\theta_{\text{old}}}(a|s))\sim\mathcal{D}}\left[\left\|\frac{\pi_{\theta_m}(a|s)}{\pi_{\theta_{\text{old}}}(a|s)}([c;r] + \gamma g_{\overline{\theta}}(s')) - g_\theta(s)\right\|^2\right]$$

   where $\theta_m$ is a frozen copy of the current parameter $\theta$.

12:     Update target gradient network parameter $\overline{\theta} \leftarrow \tau\theta + (1 - \tau)\overline{\theta}$.

13:     Update $(u, w) = (u^m, w^m)$ using the projected gradient descent:

$$(u^{m+1}, w^{m+1}) = \mathcal{P}_{K,L}\left[(u^m, w^m) - l_m(\mathbb{E}_{s\sim\mu_0}[g_\theta(s)] - [C_{th}; \mathbf{0}_K])\right].$$

14:     Schedule current learning rate of the weight $l_m$.

15:     **for** $j = 0, \cdots, U_s - 1$ **do**

16:         Sample a batch of data $\mathcal{B} \subset \mathcal{D}$.

17:         $Q_\phi \leftarrow$ **Critic Update**$(Q_\phi, Q_{\overline{\phi}}, \pi_\theta, (u^{m+1}, w^{m+1}), \mathcal{B})$

18:     **end for**

19:     Update target critic parameter $\overline{\phi} \leftarrow \tau\phi + (1 - \tau)\overline{\phi}$.

20:     $\pi_\theta \leftarrow$ **Actor Update**$(Q_\phi, \pi_\theta, \mathcal{D})$

21: **end for**

22: Return $\pi_\theta$.

---

**Algorithm 3** Critic Update

**Input**: critic $Q_\phi$, target critic $Q_{\overline{\phi}}$, actor $\pi_\theta$, weight $(u, w)$, sample batch $\mathcal{B}$

2: Update the critic parameter $\phi$ as follows:

$$
\phi \leftarrow \phi - l_c \nabla_\phi \frac{1}{|\mathcal{B}|} \sum_{(s,a,r,s') \in \mathcal{B}} \left( \sum_{k=1}^K w_k r^{(k)}(s,a) + \sum_{l=1}^L u_l c^{(l)}(s,a) \right.
$$
$$
\left. + \gamma \beta \log \mathbb{E}_{a' \sim \pi_\theta} \left[ \frac{\exp\left(Q_{\overline{\phi}}(s', a')/\beta\right)}{\pi_\theta(a'|s')} \right] - Q_\phi(s,a) \right)^2
$$

(82)

where $l_c$ is a critic learning rate.

**Output**: Updated critic $Q_\phi$

---

**Algorithm 4** Actor Update

**Input**: critic $Q_\phi$, actor $\pi_\theta$, replay buffer $\mathcal{D}$

Sample a batch of data $\mathcal{B} \subset \mathcal{D}$ and find the actor satisfying the following:

$$
\theta \leftarrow \arg\min_\theta \mathbb{E}_{s \sim \mathcal{B}} \mathbb{E}_{a \sim \pi_\theta(\cdot|s)} \left[ \beta \log \pi_\theta(a|s) - Q_\phi(s,a) \right]. \tag{83}
$$

3: **Output**: Updated actor $\pi_\theta$

---

## I.2 Environmental Details: Resource Allocation

We modified the source code of the edge computing simulator (Bae et al., 2020) uploaded to `https://github.com/sosam002/KAIST_MEC_simulator`, implemented with $N_{\text{type}} = 3$. Here, $K = N_{\text{type}}$ and $L = 1$.

At each timestep, the system observes a state containing the current length of each task queue. Based on this state, it selects a $2(N_{\text{type}} + 1)$-dimensional nonnegative continuous action $a_t = [a_e^{(1)}(t), \cdots, a_e^{(N_{\text{type}}+1)}(t), a_c^{(1)}(t), \cdots, a_c^{(N_{\text{type}}+1)}(t)]$. Here, $\{a_e^{(i)}(t)\}_{i=1}^{N_{\text{type}}}$ denotes the CPU core allocation ratios across task queues at the edge node, subject to the constraint $\sum_{i=1}^{N_{\text{type}}+1} a_e^{(i)}(t) = 1$. Similarly, $\{a_c^{(i)}(t)\}_{i=1}^{N_{\text{type}}}$ denotes the bandwidth allocation ratios at the cloud node, with the constraint $\sum_{i=1}^{N_{\text{type}}+1} a_c^{(i)}(t) = 1$.

Each state is represented by a 16-dimensional vector that captures both dynamic and static characteristics. The edge device contributes 15 dimensions, derived from three application queues, each described by five features: (1) average task arrivals over the most recent 10 timesteps, (2) task arrivals at the current timestep, (3) current queue lengths, (4) CPU utilization ratios, and (5) fixed workload values per application. The remaining dimension represents the current CPU utilization ratio of the cloud server. Among these features, the workload values per application are static, defined as fixed CPU cycles per bit, while all other dimensions vary dynamically over time.

Table 7: Parameters for Each Application Types ($K = N_{\text{type}} = 3$)

| Application | Workload | Popularity | Min Bits | Max Bits |
|---|---|---|---|---|
| SPEECH RECOGNITION | 10435 | 0.5 | 40 KB | 300 KB |
| NATURAL LANGUAGE PROCESSING | 25346 | 0.8 | 4 KB | 100 KB |
| VIRTUAL REALITY | 40305 | 0.1 | 0.1 MB | 3 MB |

Table 7 summarizes the key parameters for each application (Bae et al., 2020). The *workload* (CPU cycles/bit) indicates the computational load per application. The *popularity* represents the average arrival rate of incoming tasks modeled by a Poisson distribution. Each application's input data size follows a normal distribution, bounded between the specified *minimum* and *maximum bits*, reflecting diverse and practical scenarios.

Each episode consists of 1,000 timesteps. The total training spans 2 million timesteps, with evaluations conducted at the end of every episode, resulting in 2,000 evaluation points. An episode is run during each evaluation and the cumulative discounted sum of the $(L + K)$-dimensional vector reward is computed. These experiments were conducted using an NVIDIA TITAN X GPU (12GB) across twelve random seeds.

### I.3 UNCONSTRAINED MAX-MIN MORL ALGORITHM

---

**Algorithm 5** Gaussian-smoothing-based Max-min Algorithm for Continuous Action (Our modification from Park et al. (2024))

---

1: $\pi_\theta$: actor, $Q_\phi$: critic, $Q_{\overline{\phi}}$: target critic, $\mathcal{D}$: replay buffer, $T_{\text{init}}$: initial iteration number, $\tau$: target update ratio, $U$: main iteration number, $U_s$: gradient step for critic update, $N_s$: number of perturbed samples, $\mu$: perturbation parameter, $l_0$: initial learning rate of the weight $w$, $K$: reward dimension

2: Initialize target critic $\overline{\phi} \leftarrow \phi$ and weight $w^0 \in \Delta^K$.

3: **for** $j = 0, \cdots, T_{\text{init}} - 1$ **do**

4:     Rollout sample from $\pi_\theta$ and save it in $\mathcal{D}$. Sample a batch of data $\mathcal{B} \subset \mathcal{D}$.

5:     $Q_\phi \leftarrow$ **Critic Update**$(Q_\phi, Q_{\overline{\phi}}, \pi_\theta, w^0, \mathcal{B})$ (Algorithm 3 without the term of $\sum_{l=1}^{L} u_l c^{(l)}(s, a)$)

6:     Update target critic parameter $\overline{\phi} \leftarrow \tau\phi + (1 - \tau)\overline{\phi}$.

7:     $\pi_\theta \leftarrow$ **Actor Update**$(Q_\phi, \pi_\theta, \mathcal{D})$ (Algorithm 4)

8: **end for**

9: **for** $m = 0, 1, 2, \cdots, U - 1$ **do**

10:     Rollout sample from $\pi_\theta$ and save it in $\mathcal{D}$.

11:     Generate $N_s$ perturbed weights $\{w^m + \mu u_n^m\}_{n=1}^{N_s}$, $u_n^m \sim \mathcal{N}(0, I_K)$.

12:     Make $N_s$ copies of $Q_\phi : \{\hat{Q}_{\phi,\text{copy},n}\}_{n=1}^{N_s}$. Sample a common batch of data $\mathcal{B}_c \subset \mathcal{D}$.

13:     **for** $n = 1, \cdots, N_s$ **do**

14:         $\hat{Q}_{w^m + \mu u_n^m,\text{copy},n} \leftarrow$ **Critic Update**$(\hat{Q}_{\phi,\text{copy},n}, Q_{\overline{\phi}}, \pi_\theta, w^m + \mu u_n^m, \mathcal{B}_c)$

15:     **end for**

16:     Calculate $\hat{L}(w^m + \mu u_n^m) = \mathbb{E}_{s \sim \mu_0} \left[ \beta \log \mathbb{E}_{a \sim \pi_\theta} \left[ \frac{\exp[\hat{Q}_{w^m + \mu u_n^m,\text{copy},n}(s,a)/\beta]}{\pi_\theta(a|s)} \right] \right]$.

17:     Conduct linear regression using $\{w^m + \mu u_n^m, \hat{L}(w^m + \mu u_n^m)\}_{n=1}^{N_s}$ and calculate the linear weight $a_m$. Discard $\{\hat{Q}_{w^m + \mu u_n^m,\text{copy},n}\}_{n=1}^{N_s}$.

18:     Update $w = w^m$ using the projected gradient descent:

$$w^{m+1} = \text{proj}_{\Delta^K} \left( w^m - l_m a_m \right).$$

19:     Schedule current learning rate of the weight $l_m$.

20:     **for** $j = 0, \cdots, U_s - 1$ **do**

21:         Sample a batch of data $\mathcal{B} \subset \mathcal{D}$.

22:         $Q_\phi \leftarrow$ **Critic Update**$(Q_\phi, Q_{\overline{\phi}}, \pi_\theta, w^{m+1}, \mathcal{B})$

23:     **end for**

24:     Update target critic parameter $\overline{\phi} \leftarrow \tau\phi + (1 - \tau)\overline{\phi}$.

25:     $\pi_\theta \leftarrow$ **Actor Update**$(Q_\phi, \pi_\theta, \mathcal{D})$

26: **end for**

27: Return $\pi_\theta$.

---

## I.4 HYPERPARAMETERS FOR RESOURCE ALLOCATION

Table 8: Hyperparameters for Algorithms ($K = N_{\text{type}}$)

| Parameter | Value |
|---|---|
| **Shared** | |
| optimizer | Adam (Kingma & Ba, 2015) |
| discount ($\gamma$) | 0.99 |
| target update interval | 1 |
| target smoothing ratio ($\tau$) | 0.001 |
| gradient steps | 1 |
| reward dimension | 3 or 8 |
| max episode step | 1000 |
| replay buffer size | $2 \times 10^6$ |
| hidden layers | 2 |
| hidden units per layer | 64 |
| minibatch size | 32 |
| activation function | ReLU |
| entropy coefficient | 0.05 |
| weight learning rate | 0.01 |
| weight scheduling | $1/\sqrt{t}$ |
| **Constrained Max-min MORL (Ours)** | |
| constraint type | maximize |
| constraint dimension | 1 |
| constraint epsilon | 1.0 |
| constraint threshold | $-5.6$ |
| main learning rate | $7.5 \times 10^{-4}$ |
| gradient steps for critic update | 3 |
| gradient estimation learning rate | $1 \times 10^{-5}(N_{\text{type}} = 3), 1.25 \times 10^{-5}(N_{\text{type}} = 8)$ |
| gradient estimation steps | 1 |
| gradient target smoothing ratio | 0.001 |
| **Unconstrained Max-min MORL (Gaussian)** | |
| main learning rate | $7.5 \times 10^{-4}$ |
| perturbation $q$ learning rate | 0.073 |
| perturbation gradient steps | 1 |
| gradient steps for critic update | 3 |
| perturbation $q$-copies | 10 |
| perturbation noise std-dev | 0.01 |
| **Unconstrained Max-min MORL (ARAM)** | |
| main learning rate | $7.5 \times 10^{-4}$ |
| CI coefficient $\eta$ | 0.01 |
| MD coefficient $\lambda$ | 0.03 |
| **Max-average SAC with a Lagrangian Relaxation** | |
| constraint type | minimize |
| initial lambda | 1.0 |
| main learning rate (actor/critic) | $3 \times 10^{-4}$ |
| constraint threshold | 5.6 |
| entropy coefficient | 0.05 |
| lambda learning rate | 0.001 |
| **Unconstrained Max-average SAC** | |
| main learning rate (actor/critic) | $3 \times 10^{-4}$ |

## I.5 HYPERPARAMETERS FOR LOCOMOTION CONTROL

Table 9: Hyperparameters for Algorithms

| Parameter | Value |
|---|---|
| **Shared** | |
| optimizer | Adam (Kingma & Ba, 2015) |
| discount ($\gamma$) | 0.99 |
| target update interval | 1 |
| target smoothing ratio ($\tau$) | 0.001 |
| gradient steps | 1 |
| reward dimension | 2 |
| max episode step | 1000 |
| replay buffer size | $1 \times 10^6$ |
| hidden layers | 2 |
| hidden units per layer | 64 |
| minibatch size | 32 |
| activation function | ReLU |
| entropy coefficient | 0.05 |
| weight learning rate | 0.001 |
| weight scheduling | $1/\sqrt{t}$ |
| **Constrained Max-min MORL (Ours)** | |
| constraint type | maximize |
| constraint dimension | 1 |
| constraint epsilon | 1.0 |
| constraint threshold | $-50$ |
| main learning rate | $7.5 \times 10^{-4}$ |
| gradient steps for critic update | 3 |
| gradient estimation learning rate | $2.5 \times 10^{-5}$ |
| gradient estimation steps | 1 |
| gradient target smoothing ratio | 0.001 |
| **Unconstrained Max-min MORL (Gaussian)** | |
| main learning rate | $7.5 \times 10^{-4}$ |
| perturbation $q$ learning rate | 0.073 |
| perturbation gradient steps | 1 |
| gradient steps for critic update | 3 |
| perturbation $q$-copies | 10 |
| perturbation noise std-dev | 0.01 |
| **Unconstrained Max-min MORL (ARAM)** | |
| main learning rate | $7.5 \times 10^{-4}$ |
| CI coefficient $\eta$ | 0.2 |
| MD coefficient $\lambda$ | 0.03 |
| **Max-average SAC with a Lagrangian Relaxation** | |
| constraint type | minimize |
| initial lambda | 1.0 |
| main learning rate (actor/critic) | $3 \times 10^{-4}$ |
| constraint threshold | 50 |
| entropy coefficient | 0.05 |
| lambda learning rate | 0.001 |
| **Unconstrained Max-average SAC** | |
| main learning rate (actor/critic) | $3 \times 10^{-4}$ |

## J  TRAFFIC SIGNAL CONTROL

To further evaluate scalability, we extend our method to an environment with a larger objective space. We note that MORL benchmark environments, particularly those with more than four objectives, are still limited (Hayes et al., 2022; Park & Sung, 2025). To address this gap, we include a traffic signal control environment (Alegre, 2019) with a 16-dimensional objective vector (Park & Sung, 2025; Byeon et al., 2025), which to our knowledge represents the largest number of objectives explored in MORL to date.

| Algorithm | Cost sum ($C_{th} = 60{,}000$) | Minimum return ($\uparrow$) |
|---|---|---|
| Random | **57,702** | $-31{,}746$ |
| MA-PPO | 77,757 | $-20{,}434$ |
| MA-CPPO | **53,160** | $-26{,}410$ |
| Ours | **48,230** | $-26{,}798$ |
| Max-min GS | 72,234 | $-21{,}532$ |
| ARAM | 88,748 | $-19{,}700$ |

Table 10: Traffic signal control results over five seeds with the constraint-satisfying algorithms highlighted in **bold**

In a simulated 16-lane four-way intersection, the agent manages the traffic lights using thirty-seven-dimensional continuous traffic states. The feedback signal consists of a 16-dimensional reward vector, where each component represents the negative waiting time of a corresponding lane. Following prior work (Park & Sung, 2025; Byeon et al., 2025), we evaluate performance in an asymmetric traffic flow scenario. Our goal is to achieve fair traffic flow across all lanes while enforcing a constraint on total $CO_2$ emissions, contributing to a more sustainable traffic control system. Since this environment operates in a discrete action space, we replaced the MA-SAC and MA-SAC-L baselines with PPO-based variants (MA-PPO and MA-CPPO). MA-CPPO denotes max-average constrained PPO, which applies clipping only to the scalar reward while keeping the Lagrangian update unclipped, following Liu et al. (2019) to improve constraint satisfaction. We ran each method for 100k timesteps per seed, using five random seeds. As shown in Table 10, among the three algorithms that satisfy the constraint, both our method and MA-CPPO outperform the Random baseline in terms of max-min performance. However, our method requires less cost than MA-CPPO, demonstrating its ability to better balance constraint satisfaction and max-min fairness.

Table 11: Hyperparameters for traffic signal control environment

| Parameter | Value |
|---|---|
| **Shared** | |
| optimizer | Adam (Kingma & Ba, 2015) |
| discount ($\gamma$) | 0.99 |
| target update interval | 1 |
| target smoothing ratio ($\tau$) | 0.001 |
| gradient steps | 1 |
| reward dimension | 16 |
| total seconds per episode | 9000 |
| delta time (seconds) | 30 |
| total timesteps | $1 \times 10^5$ |
| replay buffer size | $1 \times 10^5$ |
| hidden layers | 2 |
| hidden units per layer | 64 |
| minibatch size | 32 |
| activation function | ReLU |
| entropy coefficient | 0.05 |
| weight scheduling | $1/\sqrt{t}$ |
| **Constrained Max-min MORL (Ours)** | |
| constraint type | minimize |
| constraint dimension | 1 |
| constraint epsilon | -1.0 |
| constraint threshold | $6.0 \times 10^4$ |
| main learning rate | $7.5 \times 10^{-4}$ |
| weight learning rate | 0.01 |
| gradient steps for critic update | 3 |
| gradient estimation learning rate | $1.0 \times 10^{-4}$ |
| gradient estimation steps | 1 |
| gradient target smoothing ratio | 0.001 |
| **Unconstrained Max-min MORL (Gaussian)** | |
| main learning rate | $7.5 \times 10^{-4}$ |
| weight learning rate | 0.01 |
| perturbation $q$ learning rate | 0.073 |
| perturbation gradient steps | 1 |
| gradient steps for critic update | 3 |
| perturbation $q$-copies | 20 |
| perturbation noise std-dev | 0.01 |
| **Unconstrained Max-min MORL (ARAM)** | |
| main learning rate | 0.001 |
| CI coefficient $\eta$ | 0.00202 |
| MD coefficient $\lambda$ | 0.2 |
| **Constrained Max-average PPO** | |
| main learning rate | 0.01 |
| constraint type | minimize |
| constraint threshold | $6.0 \times 10^4$ |
| constraint learning rate | 0.005 |
| **Unconstrained Max-average PPO** | |
| main learning rate | 0.01 |

## K BROADER IMPACT

In this work, we propose an algorithm for constrained MORL based on the max-min criterion. First, max-min MORL plays a critical role in promoting fairness across objectives in domains such as traffic management and resource allocation. Unfair results can lead to user dissatisfaction and, in turn, degrade overall system performance, for example, by contributing to traffic congestion (Raeis & Leon-Garcia, 2021). Second, incorporating constraints into RL is essential for the responsible development of AI systems, especially given real-world limitations on resources such as electricity, power consumption, and fossil fuels.

Our work advances the goal of **sustainable AI** by simultaneously incorporating fairness and resource constraints into decision-making. This contrasts to traditional methods that prioritize performance alone, often overlooking concerns of equity and efficient resource use. We believe our framework has the potential to make a meaningful and positive impact on the broader AI community, not only in resource allocation but also in emerging areas such as fair and safe alignment of large language models.

## L LIMITATION, FUTURE WORK, AND DISCUSSION

In this section, we discuss several limitations of our work and related future research avenues, although our method offers a promising direction for developing constrained MORL algorithms.

First, there is a lack of well-established benchmarks for MORL compared to standard RL settings (Hayes et al., 2022), and even fewer environments are specifically designed for constrained MORL. Additionally, most existing MORL environments have low-dimensional reward spaces (typically fewer than four dimensions) (Park & Sung, 2025), which limits the ability to evaluate our algorithm in high-dimensional settings. Developing practical benchmarks for both MORL and constrained MORL is therefore a critical research direction for the community.

Second, while it is common in the constrained MDP literature to assume that feasibility is ensured by appropriately chosen thresholds (Tessler et al., 2018; Ha et al., 2020), determining such thresholds, that is, setting the constraint set $\{C^{(l)}\}_{l=1}^{L}$, is non-trivial in practice outside of simple or tabular domains. Unlike trial-and-error reward design, constraint threshold design is often infeasible or unsafe due to the potential risks and costs involved. Leveraging external sources of information, such as human demonstrations or natural language descriptions, offers a promising path for setting constraint thresholds in constrained RL and MORL. Another possible approach is to infer the constraint values from expert demonstrations, commonly referred to as inverse constrained RL (Malik et al., 2021; Subramanian et al., 2024).

Third, while our resource allocation setting clearly distinguishes rewards from costs, this distinction may be ambiguous in other domains. Determining which objectives should be treated as constraints versus unconstrained rewards can be challenging. As with constraint threshold design, incorporating external guidance could help better structure constrained MORL problems.

Fourth, several constrained RL studies have explored more conservative formulations than those based on expected cumulative cost, for example, using outage probability or quantile-based constraints to manage rare but critical failures in domains such as finance or insurance (Yang et al., 2021; Jung et al., 2022). While our current framework and analysis rely on expected cumulative cost, extending it to support such conservative constraint formulations presents a valuable direction for future work.

Lastly, although we assume the convergence of the (action) value function for each weight pair $(u, w)$, it is well known that the combination of function approximation, bootstrapped updates, and off-policy learning can lead to instability and even divergence during training (Sutton & Barto, 2018; Che et al., 2024). A theoretical investigation into this so-called *deadly triad*, along with additional convergence guarantees, would further improve the robustness of our algorithm in the resource allocation experiment and broaden its applicability to other domains.

Scalarization-based methods are highly valuable, especially because of their interpretability and flexibility in expressing designer preferences. In particular, when incorporating constraints, these methods also make it straightforward to assess constraint satisfaction through the corresponding dual variables. However, linear scalarization cannot recover nonlinear or concave regions of the

Pareto frontier, potentially missing desirable trade-off solutions (Roijers et al., 2013; Hayes et al., 2022). While mixtures of convex scalarization functions can help approximate concave regions, this often requires careful tuning and may increase computational effort. Addressing the limitations of scalarization-based approaches is indeed valuable.

We also agree that, in many real-world systems, verifying smoothness analytically or designing theoretically optimal initialization is not straightforward. However, our method does not require users to prove smoothness beforehand, nor does it rely on fragile initialization. Instead, smoothness is used to (i) guarantee convergence behavior theoretically, and (ii) characterize how approximation errors propagate during weight updates. In practice, we adopt standard strategies such as entropy regularization and projection onto the simplex, which naturally stabilize learning without requiring explicit verification of smoothness. Empirically, we observe convergence in our paper even without tuning for smoothness-related parameters beyond the default hyperparameters. Regarding initialization, while local methods can be sensitive in general nonlinear optimization, we find that (i) uniform initialization on the probability simplex consistently works across tasks, and (ii) the algorithm does not require task-specific warm-starts. Finally, we note that the purpose of our theoretical results is not to imply that all assumptions will be checked analytically in practice, but rather to provide predictable behavior and guidance for practical usage of our algorithm.

