# OpenReview forum: "Constrained Multi-Objective Reinforcement Learning with Max-Min Criterion"
_ICLR.cc/2026/Conference — Submitted to ICLR 2026_

### Official Review · Reviewer_1Uxo · 2025-10-29

**Soundness:** 3
**Presentation:** 2
**Contribution:** 3
**Rating:** 6
**Confidence:** 2

**Summary:**

This paper proposes a unified framework for constrained MORL that simultaneously addresses the max–min criterion and constraint satisfaction, while also establishing a solid theoretical foundation for the framework. The authors further develop an iterative algorithm with a theoretical analysis of its convergence rate. Empirically, they demonstrate that the proposed method performs well not only in tabular settings but also on more practical tasks.

**Strengths:**

1. The proposed framework for constrained max–min MORL is impressive. In establishing its theoretical foundation, the authors effectively combine reinforcement learning and optimization techniques to derive a tractable constrained optimization formulation of the primal problem, which serves as the basis for the practical algorithm.

2. The experimental section goes beyond tabular settings and includes trials on more practical environments, showing the method’s applicability.

**Weaknesses:**

1. The proposed method appears to perform well only on small-scale MORL tasks, where the number of objectives remains limited. It would strengthen the paper to extend the evaluation to settings with a larger number of objectives, thereby demonstrating the method’s practical scalability and value.

2. Although the paper provides a theoretical analysis of convergence, it does not include learning curves to empirically illustrate convergence behavior. Including such results would make the empirical section more convincing.

**Questions:**

1. The paper mentions heterogeneous objectives. Could the authors clarify how the proposed design effectively handles these heterogeneous objectives?

---

> ### Author Response · Authors · 2025-11-21
>
> We thank the reviewer for the constructive and thoughtful feedback. In the revised paper, the modified text is highlighted in blue. **We have also updated the Related Work section to clarify that this paper is fundamentally different from prior work.**
>
> We address the reviewer’s comments below.
>
> &nbsp;
>
> **1. Additional experiment with a larger number of objectives and a recent MORL baseline**
>
> We note that benchmark environments for MORL, particularly those with more than four objectives, are still limited [1,2]. We include a traffic signal control environment with 16-dimensional objectives [2,3], which to our knowledge represents the largest number of objectives explored in MORL to date. In a simulated 16-lane four-way intersection, the agent manages the traffic lights using thirty-seven-dimensional continuous traffic states. The feedback signal consists of a 16-dimensional reward vector, where each component represents the negative waiting time of a corresponding lane. Our goal is to achieve fair traffic flow across all lanes while enforcing a constraint on total CO2 emissions. For completeness, we have included the state-of-the-art max-min MORL algorithm ARAM from the concurrent work [3] (as well as in resource allocation and locomotion control environments).
>
> As shown in the table, among the three algorithms that satisfy the constraint (highlighted in **bold**), both our method and MA-CPPO outperform the Random baseline in terms of max-min performance. (Here, MA-CPPO denotes max-average constrained PPO and GS Gaussian smoothing.) However, our method requires less cost than MA-CPPO, demonstrating its ability to better balance constraint satisfaction and max-min fairness.
>
> | Algorithm    | Cost sum (C_th = 60,000) | Minimum return (↑) |
> |--------------|------------------------------|-------------------------|
> | Random       | **57,702**         | -31,746                 |
> | MA-PPO       | 77,757         | -20,434                 |
> | MA-CPPO     | **53,160**           | -26,410          |
> | Ours     | **48,230**          | -26,798         |
> | Max-min GS   | 72,234           | -21,532             |
> | ARAM         | 88,748         | -19,700            |
>
> [1] Hayes et al., "A practical guide to multi-objective reinforcement learning and planning", 2022.
>
> [2] Park and Sung, "Reward dimension reduction for scalable multi-objective reinforcement learning", 2025.
>
> [3] Byeon et al., "Multi-Objective Reinforcement Learning with Max-Min Criterion: A Game-Theoretic Approach", 2025.
>
> **2. Analysis of learning curves for convergence**
>
> We have included the learning curve for tabular convergence in Figure 1 (Section 5.1) in the revised paper. Our method converges reliably to the optimal value, whereas the Gaussian smoothing baseline exhibits larger approximation errors and unstable learning behavior. In summary, our method is superior in accuracy as well as computation for constrained max-min optimization compared to Gaussian smoothing in tabular settings.
>
> **3. Clarification on addressing heterogeneous objectives**
>
> To clarify, our goal is not to enforce max-min fairness across fundamentally different quantities with incompatible units (e.g., queue delay vs. power consumption). Instead, our method enforces max-min fairness among homogeneous objectives while **incorporating other heterogeneous quantities as constraints rather than fairness objectives.**
>
> This design broadens the applicability of conventional unconstrained max-min MORL to practical scenarios such as fair resource allocation under system-level power limits. As shown in Section 5.2, applying max-min MORL directly in such settings without modeling the constraints leads to excessive power consumption, making it impractical despite improved fairness.
>
> We agree that the initial draft could give the impression that we perform max-min optimization over heterogeneous objectives. We have updated our Introduction section in the revised paper to clarify this point.

---

### Official Review · Reviewer_4QKe · 2025-11-01

**Soundness:** 2
**Presentation:** 2
**Contribution:** 3
**Rating:** 4
**Confidence:** 3

**Summary:**

- The authors propose an algorithm that maximizes the minimum of multiple objectives (fairness) while satisfying constraints.
- To formulate the constrained multi-objective RL (CMORL) problem as a convex optimization, they convert the problem to a dual problem using the occupancy measure.

**Strengths:**

- The authors transform the CMORL problem into a convex dual problem, thereby reducing its complexity.
- They derive the update rules for the dual variables (v, w) using value functions from standard RL frameworks, lowering the implementation difficulty.
- They prove the convergence of the proposed method.

**Weaknesses:**

- Insufficient survey of prior CMORL work.
    - The authors focus on resolving the issue that linear weights fail to reflect user preferences for heterogeneous objectives.
    - CoMOGA [1] was proposed to address the same problem, yet it is neither cited nor compared experimentally, which is a notable omission.
- The introduction needs more explanation.
    - It does not clarify how fairness leads to max-min optimization.
    - At line 54, a simple example illustrating the power of max-min optimization would be helpful.
    - Even with heterogeneous objectives, scale-invariant algorithms like CoMOGA can resolve the issue by taking uniform relative importance as user preference; the need for max-min over this alternative should be explicitly addressed.
- The claim that the proposed method can handle heterogeneous objectives is not strongly supported, as the proposed method is only evaluated on homogeneous objectives.
- Equations 4–8 are similar to derivations in prior work but lack proper citation.
    - COptiDICE [2], a prior method that similarly reformulates constrained RL in occupancy measure space, derives analogous steps during dual transformation. These should be cited.
- Updating dual variables (u, w) and primal variables (policy, value) simultaneously is not feasible, resulting in increased computational cost.
    - As shown in Equation 15 of Theorem 3.6’s proof, the optimality gap is proportional to $\epsilon$.
    - This requires fully converging the policy for fixed (u, w) before updating (u, w), and repeating the process.
    - Consequently, convergence demands far more iterations than standard RL training process.

[1] Kim, Dohyeong, et al. "Conflict-Averse Gradient Aggregation for Constrained Multi-Objective Reinforcement Learning." The Thirteenth International Conference on Learning Representations.

[2] Lee, Jongmin, et al. "COptiDICE: Offline Constrained Reinforcement Learning via Stationary Distribution Correction Estimation." International Conference on Learning Representations.

**Questions:**

- In Table 2 (tabular setting results), the optimality gap for Gaussian smoothing is large.
    - Was the smoothing factor for Gaussian smoothing set to sufficiently small?
    - If not, the comparison may not be fair.
- All objectives in the experimental environments are homogeneous.
    - If heterogeneous objectives are converted into constraints, how should appropriate thresholds be determined?
    - Setting thresholds too high would prevent meaningful performance, while setting them too low would ignore the constraints.
    - Without clear guidelines, empirically tuning thresholds introduces an additional iterative process outside the learning algorithm, increasing overall complexity despite not being reflected in the reported computational cost.
- Why was the proposed method not compared with CMORL algorithms (e.g., LP3 [1], CoMOGA) in Section 5.2?

[1] Huang, Sandy, et al. "A constrained multi-objective reinforcement learning framework." Conference on Robot Learning. PMLR, 2022.

---

> ### Author Response · Authors · 2025-11-21
>
> We thank the reviewer for the constructive and thoughtful feedback. In the revised paper, the modified text is highlighted in blue. **We have also updated the Related Work section to clarify that this paper is fundamentally different from prior work.**
>
> We address the reviewer’s comments below.
>
> &nbsp;
>
> **1. Discussion of CoMOGA in the literature and the necessity of max-min MORL algorithm**
>
> We appreciate have cited CoMOGA [1] in the revision (the last paragraph in Section 4). In short, our method and CoMOGA address different but complementary goals within constrained MORL.
>
> In unconstrained MORL, the goal is to learn an optimal policy $\pi^\* = \arg \max_{\pi} f(J(\pi))$ given a non-decreasing scalarization function $f: \mathbb{R}^K \rightarrow \mathbb{R}$. Existing MORL methods fall into two categories: (i) Single-policy methods, which directly optimize a given non-linear scalarization function $f$, yielding a single optimal policy $\pi^\*$. (ii) Multi-policy methods, which aim to learn preference-conditioned policies  $\pi^\*(\cdot | \cdot, \omega)$ to represent a set of user preferences. These two categories are complementary, as it is generally unclear how to select the preference vector $\omega$ in a multi-policy framework so that the resulting policy corresponds exactly to $\pi^*$ for a given non-linear scalarization function $f$.
>
> In our constrained formulation, the same distinction applies. Our method corresponds to the single-policy paradigm, whereas CoMOGA (and LP3) belong to the multi-policy paradigm. However, identifying which preference vector $\omega$ in the multi-policy paradigm would yield the constrained max-min optimal policy remains non-trivial. Making a forced comparison between our work and CoMOGA in the max-min MORL setting would involve training CoMOGA, sampling many preference vectors at test time, and selecting the one with the best constrained max-min score. Yet, this procedure neither guarantees max-min optimality nor is sample-efficient.
>
> In summary, the necessity of max-min MORL is supported by its complementary relationship to CoMOGA. We have clarified this point and expanded the discussion in Appendix G.
>
> [1] Kim et al, "Conflict-Averse Gradient Aggregation for Constrained Multi-Objective Reinforcement Learning", 2025.
> &nbsp;
>
> **2. Wide use of max-min optimization**
>
> *"It does not clarify how fairness leads to max-min optimization."*
>
> When solving $\max_{\pi \in \Pi} \min_{1 \leq k \leq K}  J_k(\pi) ~ (K \geq 2)$, the optimal solution typically satisfies the equalizer property, i.e., $J_1(\pi^\*)=J_2(\pi^\*)=\cdots=J_K(\pi^\*)$ assuming the resulting equality point lies on the Pareto boundary [2]. In other words, the max-min formulation selects the Pareto-optimal point that equalizes utilities across objectives. In this sense, max-min optimization is closely related to a notion of fairness.
>
> *"A simple example ... would be helpful."*
>
> As suggested, we have modified Line 54 in the revised paper to briefly mention the following example. Max-min optimization has played a central role in wireless and edge resource management, particularly in scenarios requiring fairness guarantees (e.g., [2]). A common example arises in cloud and edge computing, where incoming jobs are decomposed into multiple interdependent subtasks [3,4]. Since downstream execution cannot proceed until all prerequisite tasks are completed, the dominant performance bottleneck becomes the slowest subtask. As a result, the objective naturally becomes minimizing the maximum processing delay-equivalently, maximizing the minimum reward across subtasks when formulated as MORL. Max-min optimization aligns with this requirement, especially when job characteristics are unknown or change over time.
>
> [2] Zehavi et al., "Weighted max-min resource allocation for frequency selective channels", 2013.
>
> [3] Saifullah et al., "Parallel real-time scheduling of dags", 2014.
>
> [4] Wang et al., "Real-time scheduling of DAG tasks with arbitrary deadlines", 2019.
> &nbsp;
>
> **3. Clarification on addressing heterogeneous objectives**
>
> To clarify, our goal is not to enforce max-min fairness across fundamentally different quantities with incompatible units (e.g., queue delay vs. power consumption). Instead, our method enforces max-min fairness among homogeneous objectives while **incorporating other heterogeneous quantities as constraints rather than fairness objectives.**
>
> This design broadens the applicability of conventional unconstrained max-min MORL to practical scenarios such as fair resource allocation under system-level power limits. As shown in Section 5.2, applying max-min MORL directly in such settings without modeling the constraints leads to excessive power consumption, making it impractical despite improved fairness. We agree that the initial draft could give the impression that we perform max-min optimization over heterogeneous objectives. We have updated our Introduction section in the revised paper to clarify this point.

---

> ### Author Response · Authors · 2025-11-21
>
> **4. Discussion of COptiDICE in the literature**
>
> We thank the reviewer for the suggestion; we have cited COptiDICE [5] in the revised manuscript (Section 4). The focus of COptiDICE is on constrained single-objective RL with a scalar reward in an offline setting. Unlike our work, it does not address fairness across multiple objectives in MORL.
>
> In detail, a key difference in our convex analysis stems from the presence of the inner min operator. Applying the KKT conditions to the slack variable that enforces the min produces the simplex constraint $\sum_{k=1}^K w_k = 1$ so the dual variables $\\{ w_k \\}_{k=1}^K$ lie on the $(K-1)$-simplex. This property reduces the effective search space and simplifies our algorithm, which in turn aids the convergence analysis even when gradient updates are inexact (see Theorem 3.6).
>
> Moreover, in Proposition 3.1 we prove an upper bound on the gap between the optimal max-min value of the unregularized problem and that of the regularized problem, showing that the regularized formulation is a valid approximation of the original criterion. By contrast, [5] does not establish such an optimality gap for its setting.
>
> [5] Lee et al., "COptiDICE: Offline Constrained Reinforcement Learning via Stationary Distribution Correction Estimation", ICLR 2022.
>
> **5. Discussion on convergence and computational aspects**
>
> Because the update in Eq. (13) is a $\gamma$-contraction, the error decays exponentially with the number of inner updates $T_{\text{in}}$. We obtain $\epsilon = O(\gamma^{T_{\text{in}}})$, which decays exponentially in $T_{\text{in}}$ and is asymptotically faster than polynomial rates. In practice, when function approximation is performed with expressive neural networks, only a small number of inner-loop updates are typically sufficient to achieve good empirical performance [6]. Across all three application domains, we observed that performing three critic gradient steps per iteration was adequate.
>
> [6] Park et al., "The Max-Min Formulation of Multi-Objective Reinforcement Learning: From Theory to a Model-Free Algorithm", 2024.
>
> **6. Fine-tuning Gaussian smoothing and learning curves for convergence**
>
> We further fine-tuned the smoothing factor of Gaussian smoothing and improved its convergence performance. We have included the learning curve for tabular convergence in Figure 1 (Section 5.1) in the revised paper. Our method converges reliably to the optimal value, whereas the Gaussian smoothing baseline exhibits larger approximation errors and unstable learning behavior. In summary, our method is superior in accuracy as well as computation for constrained max-min optimization compared to Gaussian smoothing.
>
> **7. Discussion on setting constraint thresholds**
>
>
> In the constrained MDP literature, it is standard to assume feasibility through appropriately chosen thresholds (e.g., [7,8]). In many practical systems such as resource allocation in computing infrastructures or locomotion control in robotics, the constraints typically correspond to well-defined physical limitations (e.g., thermal envelope, hardware wear tolerance, safety limits). In such settings, identifying thresholds that prevent system failure is relatively natural once the system specifications are known.
>
> However, we agree that determining the constraint set $\\{ C^{(l)} \\}_{l=1}^L$ is not always straightforward. We had previously noted this issue in the Limitation and Future Work section (now Appendix L) prior to the updated revision. In other words, in scenarios without clear physical limits, determining proper constraint levels may require additional efforts. While developing principled methods for selecting constraint thresholds is beyond the scope of our work, applying such methods to our setting would be another valuable research direction. One possible approach is to infer the constraint values from expert demonstrations, commonly referred to as inverse constrained RL [9,10].
>
> [7] Tessler et al., "Reward constrained policy optimization", 2018.
>
> [8] Ha et al., "Learning to walk in the real world with minimal human effort", 2020.
>
> [9] Malik et al., "Inverse constrained reinforcement learning", 2021.
>
> [10] Subramanian et al., "Confidence Aware Inverse Constrained Reinforcement Learning", 2024.

---

### Official Review · Reviewer_4bvt · 2025-11-04

**Soundness:** 3
**Presentation:** 2
**Contribution:** 3
**Rating:** 6
**Confidence:** 3

**Summary:**

This paper rewrites the constrained max-min multi-objective RL problem as a convex occupancy-measure program. It then gives an equivalent dual with nonnegative constraint weights u and simplex weights w, making a soft value iteration operator. With this change, the gradients of the objective with respect to the $(u,w)$ are characterized. The theoretical analysis covers convergence and sample complexity.

**Strengths:**

-	The paper clearly identifies applications where the max-min criterion is important, emphasizing scenarios where standard weighted-sum scalarization cannot fully deal with.
-	In structured MOMDPs, the method gives less optimality errors than Gaussian smoothing and is easier to update.

**Weaknesses:**

-	Core validation is tabular, and the two application studies are small-scale. There lacks large continuous-control benchmark.
-	Comparisons focus on a modified Gaussian smoothing max-min method, and stronger constrained baselines such as CPO, PCPO, and recent MORL methods are missing.
-	The analysis is limited, for example, missing analyses on number and tightness of constraints, fixing and learning $w$ in the setting. Moreover, the assumption is strong, for example, relying on strict feasibility.

**Questions:**

See the weakness.

---

> ### Author Response · Authors · 2025-11-21
>
> We thank the reviewer for the constructive and thoughtful feedback. In the revised paper, the modified text is highlighted in blue. **We have also updated the Related Work section to clarify that this paper is fundamentally different from prior work.**
>
> We address the reviewer’s comments below.
>
> &nbsp;
>
> **1. Additional experiment with a larger number of objectives and a recent MORL baseline**
>
> We note that benchmark environments for MORL, particularly those with more than four objectives, are still limited [1,2]. We include a traffic signal control environment with 16-dimensional objectives [2,3], which to our knowledge represents the largest number of objectives explored in MORL to date. In a simulated 16-lane four-way intersection, the agent manages the traffic lights using thirty-seven-dimensional continuous traffic states. The feedback signal consists of a 16-dimensional reward vector, where each component represents the negative waiting time of a corresponding lane. Our goal is to achieve fair traffic flow across all lanes while enforcing a constraint on total CO2 emissions. For completeness, we have included the state-of-the-art max-min MORL algorithm ARAM from the concurrent work [3] (as well as in resource allocation and locomotion control environments).
>
> As shown in the table, among the three algorithms that satisfy the constraint (highlighted in **bold**), both our method and MA-CPPO outperform the Random baseline in terms of max-min performance. (Here, MA-CPPO denotes max-average constrained PPO and GS Gaussian smoothing.) However, our method requires less cost than MA-CPPO, demonstrating its ability to better balance constraint satisfaction and max-min fairness.
>
> | Algorithm    | Cost sum (C_th = 60,000) | Minimum return (↑) |
> |--------------|------------------------------|-------------------------|
> | Random | **57,702** | -31,746  |
> | MA-PPO  | 77,757    | -20,434 |
> | MA-CPPO| **53,160**   | -26,410 |
> | Ours | **48,230**    | -26,798 |
> | Max-min GS   | 72,234    | -21,532  |
> | ARAM | 88,748  | -19,700 |
>
> [1] Hayes et al., "A practical guide to multi-objective reinforcement learning and planning", 2022.
>
> [2] Park and Sung, "Reward dimension reduction for scalable multi-objective reinforcement learning", 2025.
>
> [3] Byeon et al., "Multi-Objective Reinforcement Learning with Max-Min Criterion: A Game-Theoretic Approach", 2025.
>
> **2. Additional ablation study**
>
> To evaluate the effect of learning the weight vectors in our algorithm, we independently disabled the learning of $u$, of $w$, and of both $(u,w)$ while initializing $u$ to a zero vector and $w=[1/K, \cdots, 1/K] \in \Delta^K$ on the simplex. The first table demonstrates that removing the learning of either weight component noticeably increases the optimal value estimation error.
>
> | Algorithm          | Optimal value error (↓) |
> |--------------------|--------------------------|
> | Ours           | **0.004**               |
> | w/o *u* update     | 0.325                   |
> | w/o *w* update     | 0.657                   |
> | w/o *(u,w)* update | 1.008                   |
>
> Next, the following table demonstrates that the performance of Gaussian smoothing (GS) is highly sensitive to the number of objectives. In our tabular setting, we fine-tuned both GS and our method at $K=L=10$, and then evaluated them at  $K=L=12$ using the same hyperparameters. As shown, the optimal value error of GS grows dramatically when increasing dimensionality, whereas our method continues to achieve near-optimal convergence. This sensitivity arises because the GS estimator requires at least $N \ge K+L+1$ perturbed Q-table copies to perform matrix inversion [4]. In this experiment, we used $N=24$, yet GS still exhibited instability when scaling from 20 to 24 objectives.
>
> | Optimal value error (↓) | K = L = 10  | K = L = 12  |
> |--------------------------|-------------|-------------|
> | Gaussian Smoothing       | 13 × 10⁻³   | 8 × 10⁴     |
> | Ours                 | **4 × 10⁻³** | **9 × 10⁻³** |
>
> [4] Park et al., "The Max-Min Formulation of Multi-Objective Reinforcement Learning: From Theory to a Model-Free Algorithm", 2024.
>
> **3. Discussion on constraint feasibility**
>
> In the constrained MDP literature, it is standard practice to assume feasibility under appropriately chosen thresholds (e.g., [5,6]). In many real-world systems, such as resource allocation in computing environments or locomotion control in robotics, constraints correspond to physical safety limits or operational boundaries (e.g., thermal constraints, hardware wear tolerance, safety envelopes). Designers select constraint levels $\\{  C^{(l)} \\}_{l=1}^L$ based on this information, and it is generally reasonable to assume they are not overly restrictive. If needed, these thresholds can be adjusted iteratively to ensure strict feasibility.
>
> [5] Tessler et al., "Reward constrained policy optimization", 2018.
>
> [6] Ha et al., "Learning to walk in the real world with minimal human effort", 2020.

---

### Official Review · Reviewer_JtJx · 2025-11-07

**Soundness:** 3
**Presentation:** 3
**Contribution:** 2
**Rating:** 4
**Confidence:** 3

**Summary:**

The work is based on max-min scalarization in MORL and does not achieve a clear improvement over the earlier version in some of the experiments. A combination of earlier work is presented that combines the max-min criterion with constraint satisfaction. Constrains are, however, treated in a similar way as the multiple objectives used here as a contribution to the loss function. A very good theoretical analysis is shown, but a critical analysis of the assumption as needed for practical applicability is not attempted.

**Strengths:**

It seems that here an aspect of MORL is considered that aims at reward shaping in the sense of finding good combinations of several rewards. This is interesting but different from other approaches and should therefore be explained more clearly in the introduction already before starting to work with the formalism.

Scalarization-based approaches can be very useful, especially if the limitations are discussed rather than removed by suitable assumptions, so that practical applicability remains an unquestionable priority of the proposed method (as promised in the abstract).

The manuscript is well prepared, and can even gain if not disconnected from other literature on MORL that  works with the concept of Pareto optimality (not mentioned here apart from App. G).

**Weaknesses:**

The manuscript is well prepared, but seem a bit disconnected from other literature on MORL because it does not use the concept of Pareto optimality which is not even mentioned (apart from App. G) although it is  the dominant concept in other work on MORL, thus reasons for a different approach should be discussed. Likewise, a reader might want to know haw the concepts of “fairness” can be related evaluation of solutions in the Pareto optimization framework. This is particularly critical here,  as fairness is emphasized several times in the manuscript, but none of the available fairness measures is discussed explicitly, so that it remains questionable this goal is actually approached.

Nevertheless, a scalarization-based approach can be useful, but if its limitations are removed from the discussion by suitable assumptions, the practical applicability of the proposed methods remains limited, even so it is promised in the abstract.

Pareto-optimization is a widely-used concept in MORL, thus reasons for a different approach need to be discussed more openly. Likewise, a reader might want to know haw the concepts of “fairness” can be related evaluation of solutions in the Pareto optimization framework. This is particularly critical here, because fairness is emphasized several times in the manuscript, but none of the available fairness measures is discussed explicitly, so that it remains questionable this goal is actually approached.

The examples could be more indicative if not only performance is studied, but also the particular advantages are discussed that should become relevant “especially when handling heterogeneous objectives or incorporating constraints”.

In addition to its intended message, Table 1 also implies that the work is combining to existing threads rather than presenting a conceptual progress. Considering also the some of the proofs do not change  or which are similar to the earlier work because soft constraints are considered here that are largely complying with the multi-objective framework. The proofs included here are similar in complexity to earlier work, i.e. largely formal and little effort has been taken to increase relevance for practical applications where e.g. smoothness is often not easily decidable unless the problem is already solved or where the effort to realize local methods by suitable initialization of sometimes comparable to an ad-hoc solution of the problem.

**Questions:**

The parameter $\beta$ occurs in equ. 1 in a role analogous to a temperature, whereas in physics $\beta$ typically used to denote an inverse temperature. As $\beta$ occurs here mostly as inverse,would it be possible to avoid unnecessary confusion and to consider using a parameter $\frac{1}{\beta}$ instead of $\beta$?

What is the justification for introducing the dimension $L$ in Section 2.?

Would it be possible to abbreviate the word “constrained” by “constr.” rather than by “const.”?

---

> ### Author Response · Authors · 2025-11-21
>
> We thank the reviewer for the constructive and thoughtful feedback. In the revised paper, the modified text is highlighted in blue. **We have also updated the Related Work section to clarify that this paper is fundamentally different from prior work.**
>
> We address the reviewer’s comments below.
>
> &nbsp;
>
> **1. Connection of max-min MORL to the concept of Pareto boundary**
>
> We thank the reviewer for this insightful comment. As highlighted in prior work [1,2], the max-min fairness criterion has a well-established relationship to the Pareto boundary. When solving  $\max_{\pi \in \Pi} \min_{1 \leq k \leq K}  J_k(\pi) ~ (K \geq 2)$, the optimal solution $\pi^\*$ typically satisfies the equalizer property, i.e., $J_1(\pi^\*)=J_2(\pi^\*)=\cdots=J_K(\pi^\*)$
> assuming the resulting equality point lies on the Pareto boundary [1]. In other words, the max-min formulation selects the Pareto-optimal point that equalizes utilities across objectives.
>
> Furthermore, [1] shows that a weighted max-min formulation (i.e., $\max_\pi \min_k \alpha_k J_k(\pi)$ with $\\{ \alpha_k \\}_{k=1}^K$ in our setting) can be used to approximate the convex portion of the Pareto boundary by varying the weights $(\alpha_1,\cdots,\alpha_K)$. For example, when $K=2$, the optimal solution satisfies $\alpha_1 J_1(\pi^\*) = \alpha_2 J_2(\pi^\*)$, which allows exploring the boundary by adjusting the ratio $\frac{\alpha_1}{\alpha_2} = \frac{J_2(\pi^\*)}{J_1(\pi^\*)}$. By sweeping weight configurations and collecting the resulting solutions, one can reconstruct an approximation of the convex region of the Pareto boundary. (For further intuition, we refer the reviewer to Fig. 1 in [2], which illustrate how weighted max-min solutions traverse the boundary.)
>
> [1] Zehavi et al., "Weighted max-min resource allocation for frequency selective channels", 2013.
>
> [2] Park et al., "The Max-Min Formulation of Multi-Objective Reinforcement Learning: From Theory to a Model-Free Algorithm", 2024.
> &nbsp;
>
> **2. Discussion on scalarization-based approaches**
>
> We also appreciate this comment. We agree that scalarization-based methods are highly valuable, especially because of their interpretability and flexibility in expressing designer preferences. In particular, when incorporating constraints, these methods also make it straightforward to assess constraint satisfaction through the corresponding dual variables. However, linear scalarization cannot recover nonlinear or concave regions of the Pareto frontier, potentially missing desirable trade-off solutions [3,4]. While mixtures of convex scalarization functions can help approximate concave regions, this often requires careful tuning and may increase computational effort. Addressing the limitations of scalarization-based approaches is indeed valuable. We have added this discussion in Appendix L.
>
> [3] Roijers et al., "A survey of multiobjective sequential decision-making", 2013.
>
> [4] Hayes et al., "A practical guide to multi-objective reinforcement learning and planning", 2022.
> &nbsp;
>
> **3. Indicative example of max-min optimization incorporating constraints**
>
> Max-min optimization plays a key role in wireless communication and edge resource management, particularly when fairness across competing users or tasks is required (e.g., [5]). A representative example arises in cloud and edge computing, where a job is decomposed into interdependent subtasks [6,7]. Because downstream execution cannot proceed until all prerequisite tasks complete, overall progress is limited by the slowest subtask. In this context, the appropriate objective is improving the worst-performing component by effectively maximizing the minimum reward across subtasks, which aligns naturally with the max-min formulation.
>
> However, optimizing max-min fairness in isolation may violate operational constraints. In real deployments, power consumption is not simply an auxiliary cost term but it represents a hard physical constraint. Sustained high utilization can trigger thermal throttling, shorten device lifespan, and lead to downtime or additional cooling and maintenance costs [8,9]. Therefore, an effective method must not only ensure fairness but must also respect system-level feasibility.
>
> Our framework addresses this gap by jointly enforcing fairness and constraint satisfaction. The proposed method adapts to heterogeneous quantities by treating comparable objectives (e.g., task delays) within the max-min criterion, while incorporating fundamentally different physical quantities such as power consumption as explicit constraints.
>
> [5] Zehavi et al., "Weighted max-min resource allocation for frequency selective channels", 2013.
>
> [6] Saifullah et al., "Parallel real-time scheduling of dags", 2014.
>
> [7] Wang et al., "Real-time scheduling of DAG tasks with arbitrary deadlines", 2019.
>
> [8] Chen et al., "Cooling-aware resource allocation and load management for mobile edge computing systems", 2020.
>
> [9] Jiang et al., "Energy aware edge computing: A survey", 2020.

---

> ### Author Response · Authors · 2025-11-21
>
> **4. Novelty in our theoretical analysis**
>
> We respectfully disagree that our theoretical effort  is  similar in complexity to earlier work. We reformulate our problem as a convex program using occupancy measures and then derive another convex program equivalent to the dual problem, which serves as the basis for our MORL algorithm. Although [10] also leverages convex analysis with occupancy measures, its focus is on constrained single-objective RL with a scalar reward (i.e., $K=1$) in an offline setting. Unlike our work, it does not address fairness across multiple objectives in MORL settings.
>
> In detail, a key difference in our convex analysis stems from the presence of the inner min operator. Applying the KKT conditions to the slack variable that enforces the min produces the simplex constraint $\sum_{k=1}^K w_k = 1$ so the dual variables $\\{ w_k \\}_{k=1}^K$ lie on the $(K-1)$-simplex. This property reduces the effective search space and simplifies our algorithm, which in turn aids the convergence analysis even when gradient updates are inexact (see Theorem 3.6).
>
> Moreover, in Proposition 3.1 we prove an upper bound on the gap between the optimal max-min value of the unregularized problem and that of the regularized problem, showing that the regularized formulation is a valid approximation of the original criterion. By contrast, [1] does not establish such an optimality gap for its setting.
>
> [10] Lee et al., "COptiDICE: Offline Constrained Reinforcement Learning via Stationary Distribution Correction Estimation", ICLR 2022.
>
> **5. Discussion on smoothness assumptions and initialization**
>
> We thank the reviewer for raising this discussion point. We agree that, in many real-world systems, verifying smoothness analytically or designing theoretically optimal initialization is not straightforward. However, **our method does not require users to prove smoothness beforehand, nor does it rely on fragile initialization.** Instead, smoothness is used to (i) guarantee convergence behavior theoretically, and (ii) characterize how approximation errors propagate during weight updates.
>
> In practice, we adopt standard  strategies such as entropy regularization  and projection onto the simplex, which naturally stabilize learning without requiring explicit verification of smoothness. Empirically, we observe convergence in our paper even without tuning for smoothness-related parameters beyond the default hyperparameters.
>
> Regarding initialization, while local methods can be sensitive in general nonlinear optimization, we find that (i) uniform initialization on the probability simplex consistently works across tasks, and (ii) the algorithm does not require task-specific warm-starts.
>
> Finally, we note that the purpose of our theoretical results is not to imply that all assumptions will be checked analytically in practice, but rather to provide predictable behavior and guidance for practical usage of our algorithm. We have added a short discussion clarifying this point in the revision (the last paragraph of Appendix L).
>
> **6. Regarding questions**
>
> As suggested, we have updated const. to constr. in Table 1. We also plan to update the notation for $\beta$; however, to avoid confusion during the discussion phase, we will temporarily keep it as-is and revise it in the final version.
>
> To clarify, our goal is not to enforce max-min fairness across fundamentally different quantities with incompatible units (e.g., queue delay vs. power consumption). Rather, our method enforces max-min fairness among homogeneous objectives, while **incorporating other heterogeneous quantities as constraints rather than fairness objectives.** This design extends traditional unconstrained max-min MORL to more realistic settings where fairness and operational safety must coexist, such as resource allocation under power, safety, or hardware limits, which motivates the introduction of the constraint dimension $L$.

---

### Official Review · Reviewer_VYpr · 2025-11-09

**Soundness:** 2
**Presentation:** 2
**Contribution:** 1
**Rating:** 2
**Confidence:** 3

**Summary:**

This paper studies the problem of multi-objective reinforcement learning (MORL) with a max-min objective. Given K + L objectives, L of which must obey some constraints, they define an entropy regularized objective to set up a convex optimization problem that they then solve with mirror descent.

**Strengths:**

Constrained MORL is an important problem and the empirical evaluation of the results in this paper suggest their algorithm has practical application to, e.g., edge computing resource allocation.

**Weaknesses:**

This work omits several related works that I believe imply the results in this paper [1,2,3]. The main difference compared to [2] appears to be the presence of constraints, but why can’t these can be handled as in [1]? Moreover, how do the results in this work improve upon [1]?

The max-min objective is treated as a unique challenge for MORL, but [1] already shows how to reduce max-min fairness to general constrained RL, where the minimum reward for any objective is constrained to be above some alpha, the optimal value of which can then be identified through binary search. Moreover, the authors claim Park et al [4] is the only prior work to address max-min RL, which is false. [1,2,5] all address this problem explicitly.

[1] “Intersectional Fairness in Reinforcement Learning with Large State and Constraint Spaces” Eaton, Hussing, Kearns, Roth, Sengupta, Sorrell

[2] “Multi-Objective Reinforcement Learning with Max-Min Criterion: A Game-Theoretic Approach” Byeon, Park, Chae, Leshem

[3] “Reinforcement learning with convex constraints” Miryoosefi, Brantley, Daume, Dudik, Shapire

[4] “The max-min formulation of multi-objective reinforcement learning: From theory to a model-free algorithm” Park et al

[5] “On welfare-centric fair reinforcement learning”  Cousins, Asadi, Lobo, Littman

**Questions:**

Please see comments in weaknesses.

**Details Of Ethics Concerns:**

I could be very wrong about my suspicions here due to the anonymity of submissions. This paper seems extremely related to https://arxiv.org/pdf/2510.20235, which I believe was recently accepted to NeurIPS2025. Given the date of the arXiv submission and the general challenge of staying up to date on literature, this would not constitute an ethics issue in my eyes, except that this paper reads very similarly to the style of this prior work, has many of the same issues regarding citation of other related prior work, and I suspect likely shares authors with this work. If that is the case, they should certainly be citing their own prior work, as well as the other papers addressing min max MORL that are cited in their prior work, since they are clearly aware of them.

---

> ### Author Response · Authors · 2025-11-17
>
> We thank the reviewer for suggesting related works and would like to apologize for being not careful enough to include many related works. The reason that we did not include many max-min multi-objective reinforcement learning (MORL) works in this paper is that the focus of this paper is how to incorporate constraints on max-min MORL, not how to solve max-min MORL itself. That is why we rather include many works of constrained MDPs in Section 4 Related Work.  As the reviewer mentioned, we believe that max-min MORL itself is already handled by many previous works.
>
> Please note that the above five papers (suggested by the reviewer) basically consider max-min MORL itself, NOT constrained max-min MORL.  [1] considers maximizing the total return summed over all groups under constraint on each group's return above threshold. [2] considers maximizing the minimum return over reward dimensions. [3] considers minimizing the distrance of average return vector to some constraint set (their eq. (4)) and converted this problem as a min-max problem (their eq. (6)). [4] also considers maximizing the minimum return over reward dimensions. [5] considers maximizing a well-fair function of returns of multiple groups. When the well-fair function is minimum, it becomes a max-min MORL again.
>
> For clarity of explanation of difference between constrained max-min MORL (in this paper) and max-min MORL, let us consider the following simple case. We have optimization variable x and have four functions f(x), g(x), h(x) and l(x). By constrained max-min MORL, we mean the following problem:
>
> $$\max_x \min [f(x), g(x)]  ~~~such~ that~~~ h(x) >= a_1 ~~ and ~~ l(x) >= a_2  ~~~ (Eq. A)$$
>
> On the other hand, max-min MORL means
>
> $$\max_x \min [ f(x), g(x), h(x), l(x) ]~~~(Eq. B)$$
>
> Note that these two problems are different. Problem Eq.A is not easily converted into the form of Problem Eq. B. The existing results in Eq. B-type MORL do not imply the solution to Eq. A-type MORL.  As the reviewer mentioned and Paper [1] states just below their eq. (2), Problem Eq. B can be converted as a constrained problem using a slack variable, i.e.,
>
> $$\max_{x,\alpha} ~ \alpha ~~~ such ~ that ~ f(x) >= \alpha, ~~ g(x) >= \alpha,~~ h(x)>=\alpha, ~~ l(x) >= \alpha.  ~~~ (Eq.C)$$
>
> But, if we apply such slack variable technique to Problem Eq. A, then we have
>
> $$\max_{x,t} ~ t ~~~ such ~ that ~ f(x) >= t, ~~ g(x)>= t, ~~ h(x) >= a_1, ~~ l(x) >= a_2. ~~~ (Eq.D)$$
>
> Note that Eq. C and Eq. D are different problems. In Eq. D, the slack variable t is not in all constraints but only in the first two. So, the conventional max-min MORL approach to pick the worst term (via dual variable update) and update x for the problem for given dual variable does not work. Basically, Reviewer's Ref. [1] falls into this algorithm category. Instead, we need a separate approach. In this paper, we consider how to solve Eq.D-like MORL problem. We provide the cost function of dual variables, prove its smoothness and the Lipschitz continuity of its gradient, validating projected gradient descent-style solution combined with soft Q-learning style policy update, and further proved the proposed algorithm's geometric convergence.
>
> Hence, **this paper is totally different from the previous works including Reviewer-mentioned Reference** [2] at https://arxiv.org/pdf/2510.20235
>
> Although the focus of this paper is handling constraint on max-min MORL, **we have updated the Related Work section to offer an expanded discussion of max-min MORL itself** because this work is generalization of max-min MORL in the end. We apologize again.
>
> We believe that there is definite contribution in this paper, proposing a way to handle Eq.D-type MORL problems and providing a unified framework to constrained and unconstrained max-min MORL. We appreciate if the reviewer would reconsider the evaluation of our work, recognizing the problem difference.
>
> One further comment is that we think that there is a fallacy in Ref.[1] by Eaton et al. They say the solution to Problem Eq. (1) in their paper is a member of the set of minimax solution to Problem Eq. (2) of their paper. We do not think the solution of Eq. (1) is a solution to Eq. (2).  For simplicity, consider the following example:
> We have two functions f(x), g(x)  and constraints f(x) >= 1, g(x) >=1. All functions are defined on domain {x1,x2} and the functions are given f(x1)=10, g(x1)=1, f(x2)=2, g(x2)=2.
>
> Problem 1: max over x [f(x)+ g(x)] such that f(x) >=1, g(x) >=1.
>
> Problem 2: max over x min{f(x), g(x)}.
>
> All functions satisfy constraints.
>
> At x1, f+g = 11
>
> At x2, f+g = 4
>
> So, Problem 1 will choose x1 as solution.  On the other hand,
>
> At x1, min [f,g] = 1
>
> At x2, min [f,g] = 2
>
> So, Problem 2 will choose x2 as solution.
> Therefore we doubt their claim that the solution of Eq. (1) of Paper [1] is a solution to Eq. (2) of Paper [1], although this is not a main part of discussion.
>
> Thank you very much

---

### Author Response · Authors · 2025-11-17

Dear Area Chair and All Reviewers of Paper 19386

Because Reviewer VYpr raised an ethics issue on our submission, we would like to clarify that the problem considered in our submission is different from existing max-min MORL works including the reviewer-mentioned arXiv paper (his or her reference [2],   “Multi-Objective Reinforcement Learning with Max-Min Criterion: A Game-Theoretic Approach” Byeon, Park, Chae, Leshem).

We explained this in detail in our response to Reviewer VYpr. Please read our response. We believe that our submission indeed extends conventional max-min MORL to the case with further constraints and provides a unified framework for constrained and unconstrained max-min MORL, which we believe meaningful.

We will provide our response to other reviewers shortly.

Thank you very much.

---

### Comment · Area_Chair_M4Bn · 2025-11-25

Dear Reviewers,

This is a gentle reminder to please take a moment to review the authors’ rebuttal for the manuscript currently under your evaluation. Your timely feedback will help us proceed with the next steps in the review process.

Thank you for your time and assistance.

Best regards,
AC

---

### Author Response · Authors · 2025-11-29

Dear Area Chair,

Thank you for serving as the Area Chair for this submission.

We would like to provide a summary of the major comments of reviewers and our key revisions to address the reviewers’ feedback.

&nbsp;

**1. Clarification of Ethics Issue Raised by Reviewer VYpr and Expanded Related Work section**

Reviewer VYpr raised an ethics issue concerning our submission. As we explained in the rebuttal to Reviewer VYpr, our submission and his or her reference [2] consider different problems, and we sent an email to program chairs clarifying this issue.  Reviewer VYpr conjectured that our result is directly derivable from existing max-min MORL works but the reviewer did not provide any detailed explanation or derivation on how this is possible.  We think that this claim is not true. The result in our submission (constrained max-min MORL) is not directly derived from existing max-min MORL results. Please see our response to Reviewer VYpr.

Regarding related works, we extensively updated the Related Work section to include additional MORL studies referenced by the reviewers and expanded the discussion on unconstrained max-min MORL to better position our contribution within the broader context, as this work generalizes that base setting. The main focus of our paper is incorporating constraints into max-min MORL, rather than solving unconstrained max-min MORL itself. Our theoretical framework is derived from a distinct formulation, including our convergence analysis.


**2. Constraints versus objectives**

We clarified that our approach does not attempt to enforce fairness across heterogeneous quantities with incompatible units (e.g., queue delay vs. power consumption in resource allocation settings). Instead, our method enforces max-min fairness among homogeneous objectives while treating heterogeneous quantities as constraints. This design broadens the applicability of conventional unconstrained max-min MORL to practical scenarios such as fair resource allocation under system-level power limits.


**3. Added a large-scale MORL benchmark**

We incorporated a traffic signal control environment with 16 objective dimensions in the revised manuscript (Appendix J). Because existing MORL benchmark environments, particularly those with more than four objectives, are very limited, we believe this addition strengthens the empirical evaluation.


**4. Finally, we addressed all remaining technical and editorial comments and responded to each reviewer.**

&nbsp;

We believe that our work meaningfully extends max-min MORL to constrained max-min MORL and provides an effective solution that is applicable to diverse practical problem settings.

We appreciate your time and effort for the community.

Sincerely,

Authors of Paper 19386

---

### Meta-Review · Area_Chair_Ez3V · 2025-12-26

**Summary:**

The most critical concern was whether the proposed method constituted a genuinely new contribution beyond existing max–min MORL and constrained RL formulations. Several reviewers argued that related work (e.g., max–min MORL, constrained RL via slack variables, and occupancy-measure dual methods) appeared to imply or subsume the proposed approach, and that relevant literature was initially under-cited. It was also noted by the reviewers that the manuscript lacked sufficient connection to standard MORL concepts, such as Pareto optimality and established fairness notions. There was also confusion about whether the method claimed to enforce fairness across heterogeneous objectives, which raised concerns about conceptual clarity. Also some concern about the limited experimental results were raised. Overall, while multiple reviewers acknowledged solid theoretical analysis and soundness, the paper initially faced skepticism regarding incremental novelty and positioning, resulting in mixed but overall borderline-to-negative recommendations.

**Reviewer Concerns:**

The authors provided a clearer distinction between unconstrained max–min MORL and their constrained formulation, and expanded the related work section. These efforts improved clarity and presentation. The inclusion of a higher-dimensional MORL benchmark, learning curves, and ablation studies addressed some requests for stronger empirical support. The authors offered a plausible explanation distinguishing their work from the cited concurrent paper, and no further action was required on ethics grounds.

Despite the rebuttal, it is not convincing that the proposed formulation and algorithm constitute a clear conceptual or methodological advance over existing max–min MORL or constrained RL approaches. Also, the additional experimental results are still rather simple.

**Reviewer Scores:**

Reviewer VYpr (Initial: 2 – Reject)
While some clarifications were acknowledged, the reviewer would likely remain at reject or marginal reject, as concerns about novelty and overlap with prior work were only partially alleviated.

Reviewer JtJx (Initial: 4 – Marginal Reject)
May acknowledge improved exposition but likely remain below the acceptance threshold, given persistent concerns about conceptual positioning and practical relevance.

Reviewer 4QKe (Initial: 4 – Marginal Reject)
Could modestly soften their stance but would likely remain borderline reject, particularly due to missing empirical comparisons and unresolved questions about heterogeneity and constraints.

Reviewers 4bvt and 1Uxo (Initial: 6 – Marginal Accept)
These reviewers were more positive but expressed ambivalence and explicitly noted they would not object to rejection. Their scores would likely remain unchanged rather than increase.

---

### Decision · Program_Chairs · 2026-01-26

Reject